# Loss of Zfp335 triggers cGAS/STING-dependent apoptosis of post-β selection thymocytes

Jeremy J. Ratiu[1] ✉, William E. Barclay[1], Elliot Lin[1], Qun Wang[1], Sebastian Wellford[1], Naren Mehta[1], Melissa J. Harnois[1], Devon DiPalma[1], Sumedha Roy[1], Alejandra V. Contreras[2], Mari L. Shinohara[1,3], David Wiest[2] & Yuan Zhuang[1]

Production of a functional peripheral T cell compartment typically involves massive expansion of the bone marrow progenitors that seed the thymus. There are two main phases of expansion during T cell development, following T lineage commitment of double-negative (DN) 2 cells and after successful rearrangement and selection for functional TCRβ chains in DN3 thymocytes, which promotes the transition of DN4 cells to the DP stage. The signals driving the expansion of DN2 thymocytes are well studied. However, factors regulating the proliferation and survival of DN4 cells remain poorly understood. Here, we uncover an unexpected link between the transcription factor Zfp335 and control of cGAS/STING-dependent cell death in post-β-selection DN4 thymocytes. Zfp335 controls survival by sustaining expression of Ankle2, which suppresses cGAS/STING-dependent cell death. Together, this study identifies Zfp335 as a key transcription factor regulating the survival of proliferating post-β-selection thymocytes and demonstrates a key role for the cGAS/STING pathway in driving apoptosis of developing T cells.

The development of a large number of T cells with a clonally acquired T-cell receptor (TCR) in the thymus demands a small number of bone-marrow-derived progenitors to undergo vigorous expansion prior to each of the sequentially ordered TCR gene rearrangement events. The first major expansion occurs immediately upon T lineage commitment at the double-negative (DN2) stage prior to the rearrangement of any TCR gene[1–4]. The expanded T-cell progenitors enter the DN3 stage where rearrangement at the TCRβ, γ, δ gene loci become permissive[5,6]. In the post-natal thymus, the majority of DN3 cells differentiate into αβT cells due to the generation of a productively rearranged TCRβ chain. Post-β-selection DN3 cells then progress to the DN4 stage, where the second phase of expansion occurs, typically involving several rounds of rapid proliferation over the course of 2–3 days in mice[7,8]. The expansion of TCRβ-positive cells results in the generation of the post-mitotic double-positive (DP) cells, which constitute 90% of all

thymocytes in post-natal mice and humans[8]. DP cells undergo TCRα gene rearrangement and selection. Ultimately, these processes result in ~1% of all cells generated within the thymus surviving and contributing to the peripheral T-cell pool[9]. Therefore, the expansion of post-β-selection DN4 cells prior to TCRα gene rearrangement and TCR selection represents a critical amplifier to control the output of αβT cells from the thymus.

While most stages of T-cell development have been subjected to extensive genetic and functional characterization, the post-β-selection proliferative phase remains less well understood. Previous studies have shown that proliferation but not survival of DN4 cells is dependent on IL-7R signaling, which functions to repress *Bcl6* expression[10]. Similarly, proliferation during this stage of development also requires the combined activities of NOTCH and pre-TCR signaling[11–14]. This effect is in part the result of induction of Fbxl1 and Fbxl12, which

[1]Duke University, Department of Immunology, Durham, NC 27710, USA. [2]Fox Chase Cancer Center, Blood Cell Development and Function Program, Philadelphia, PA 19111, USA. [3]Duke University, Department of Molecular Genetics and Microbiology, Durham, NC 27710, USA. ✉e-mail: Jeremy.Ratiu@duke.edu

induce polyubiquitination and proteasomal degradation of Cdkn1b, thereby ensuring proper cell cycle progression and proliferation[15]. Survival of proliferating post-β-selection thymocytes has been found to require the expression of the chromatin-associated protein yin yang 1 (Yy1), the absence of which drives p53-dependent apoptosis[16]. Animal models exploring cell death during T-cell development have repeatedly shown that thymocyte apoptosis, including among DN4 cells, is largely driven by activities of proapoptotic Bcl2-family proteins[17–21]. Pathways controlling the survival and death of early proliferating thymocytes upstream of the Bcl2 family remain largely unexplored.

Underpinning the fate decisions of thymocytes are vast transcriptional networks that coordinate the intricate changes and checkpoint traversals required for proper development[22]. Numerous transcription factors function at different stages to coordinate the complex cellular changes involved in T-cell development. One transcription factor family of particular importance is the basic helix-loop-helix E proteins, which include E2A, HEB, and E2-2. In developing T cells, activities of the E2A and HEB have been shown to regulate nearly all stages of thymopoiesis[23,24]. These E proteins play critical roles in enforcing the β-selection checkpoint by promoting expression of *Rag1/2*[25] and *pre-Tα*[26], activation of the TCRβ[27], TCRγ, and TCRδ loci[28], and preventing DN cells lacking a functional TCRβ chain from progressing to the DP stage[29,30]. Additionally, E-protein activity has been shown to enforce early T-cell lineage commitment[31] and promote survival of post-β-selection DP thymocytes undergoing TCRα recombination[32]. Together, the combined activities of E proteins play critical and indispensable roles in the establishment of a functional T-cell compartment. However, due to the widespread binding of these factors throughout the genome of developing thymocytes, our understanding of their roles in development is far from complete.

The cGAS/STING pathway functions to sense cytosolic DNA and initiate innate immune responses[33]. Cyclic GMP-AMP (cGAMP) synthase (cGAS) recognizes dsDNA, typically of foreign origin, catalyzing the generation of the cyclic dinucleotide (CDN) second messenger cGAMP which in turn drives STING activation and downstream signaling[34]. The cGAS/STING pathway is best known for its functions in nonimmune and innate immune cells such as macrophage and dendritic cells in the context of viral or bacterial infections. In these contexts, activation of the pathway typically results in the production of type-I interferons and other pro-inflammatory mediators. Recent work has shown that the cGAS/STING pathway is also highly active but functionally divergent within T cells, primarily driving type-I interferon-independent responses and apoptosis[35–38]. Under steady-state conditions, the cGAS/STING pathway plays a minimal role in T-cell development, as evidenced by normal thymic T-cell subset proportions and overall thymus size in cGAS or STING-deficient C57/BL6 mice[37]. However, it remains to be determined whether the cGAS/STING pathway plays a role in sensing and responding to cell-intrinsic stresses during thymic T-cell development.

In this study, we show that loss of Zinc finger transcription factor 335 (Zfp335) triggered cGAS/STING-mediated apoptosis among proliferating DN4 cells. Mechanistically, Zfp335 functions to suppress cGAS/STING activation by promoting Ankle2 expression, which, in turn, regulates the cGAS inhibitor Baf. The importance of cGAS/STING pathway among DN4 thymocytes is further demonstrated by their sensitivity to STING agonists and STING-mediated cell death in wild-type mice. Thus, we have uncovered a role for the cGAS/STING pathway in regulating thymic T-cell development and identified the Zfp335/Ankle2/Baf axis as an important transcriptional network functioning to regulate cGAS/STING activity.

## Results

### Zfp335, an E-protein target, is critical for T-cell development
E-protein family transcription factors are indispensable regulators of nearly every stage of T-cell development[4,22,27,29,32,39–42]. E proteins control complex transcriptional networks, which remain incompletely understood. To gain deeper insight into mechanisms by which E proteins regulate T-cell development, we previously performed E2A ChIP-seq to identify the genome-wide binding sites during T-cell development[42]. We identified Zfp335 as an E-protein target during T-cell development (Fig. S1a). Analysis of published data showed E-protein-deficient thymocytes exhibit significantly reduced *Zfp335* expression (Fig. S1b)[41]. Additionally, *Zfp335* is ubiquitously expressed among thymocytes (Fig. S1c), suggesting that it may play an important role throughout T-cell development. Since germline deletion of *Zfp335* is nonviable[43], we utilized a conditional deletion model in which Cre expression is controlled by the E8$_{III}$ enhancer of *Cd8a* (E8$_{III}$-cre) to allow functional assessment of Zfp335 in post-β-selection thymocytes[44]. There are conflicting reports regarding the deletion kinetics for this Cre[44,45], therefore, we began by assessing its activity across T-cell development in our system (Fig. S1d, e). Consistent with Dashtsoodol et al., we found E8$_{III}$-cre is highly active immediately upon entry into DN3a with no recombination activity evident in the preceding DN2 stage. However, deletion does not appear to be complete until the DP stage.

We subsequently assessed *Zfp335*$^{fl/fl}$ E8$_{III}$-cre (Zfp335cKO) mice for thymic T-cell development. Deletion of *Zfp335* led to a significant reduction in total thymic cellularity (Fig. 1a, b). This reduction in thymic cellularity is likely due to defects in the αβ lineage as numbers of γδ T cells were not altered (Fig. 1c, d). Assessment of developmental stages revealed the reduction in thymocyte numbers of Zfp335cKO mice begins at the DN4 stage (Fig. 1e–i).

Examination of the peripheral T-cell compartment revealed significantly reduced numbers of splenic T cells in Zfp335cKO mice (Fig. S2a–g). A previous study identified the hypomorphic *Zfp335*$^{bloto}$ allele as the causative mutation in a unique form of T lymphopenia[46]. Like *Zfp335*$^{bloto}$ mice, we found that peripheral T cells in Zfp335cKO mice were almost exclusively of an effector or memory phenotype (Fig. S2h–k) suggesting these mice also exhibit a similar defect in the establishment of the naïve T-cell compartment.

To determine the transcriptional changes resulting from the loss of Zfp335 we performed RNA-seq on Zfp335cKO DP thymocytes. DP cells were used as they were the first population exhibiting complete deletion (Fig. S1e). We found that loss of Zfp335 results in differential expression of 327 genes (113 down, 214 up; Fig. 1j, k). Among the 161 Zfp335 ChIP-seq targets identified in thymocytes[46], 34 were downregulated in Zfp335cKO mice (Fig. 1k). No Zfp335 target genes were upregulated in Zfp335cKO samples (Fig. 1k), corroborating previous findings that Zfp335 primarily functions as a transcriptional activator[43,46]. Consistent with transcriptomic analyses of *Zfp335*$^{bloto}$ mice[46], geneset enrichment analysis (GSEA) revealed significant enrichment for type-I and type II interferon signaling and P53 signaling pathways in Zfp335cKO DP cells (Fig. 1l). Together, these findings identify Zfp335 as a key transcription factor regulating T-cell development.

### Loss of Zfp335 in DN3 thymocytes does not impair β-selection
Zfp335 deletion results in reduced cell numbers beginning at the DN4 stage, raising the possibility that the inability to rearrange the TCRβ locus could be responsible. Consequently, we assessed TCRβ rearrangement in DN3 and DN4 thymocytes by intracellular staining. The frequency of icTCRβ$^+$ cells among Zfp335cKO DN3 and DN4 subsets was comparable to that of WT (Fig. S3a–c). Therefore, TCRβ rearrangement and subsequent pre-TCR expression are unimpaired in Zfp335cKO mice.

In addition to pre-TCR expression, to successfully traverse the β-selection checkpoint, pre-TCR signals are required for survival, release from cell cycle arrest, and progression to DP[47]. CD27 surface expression is increased by pre-TCR signals in DN3 thymocytes[48].

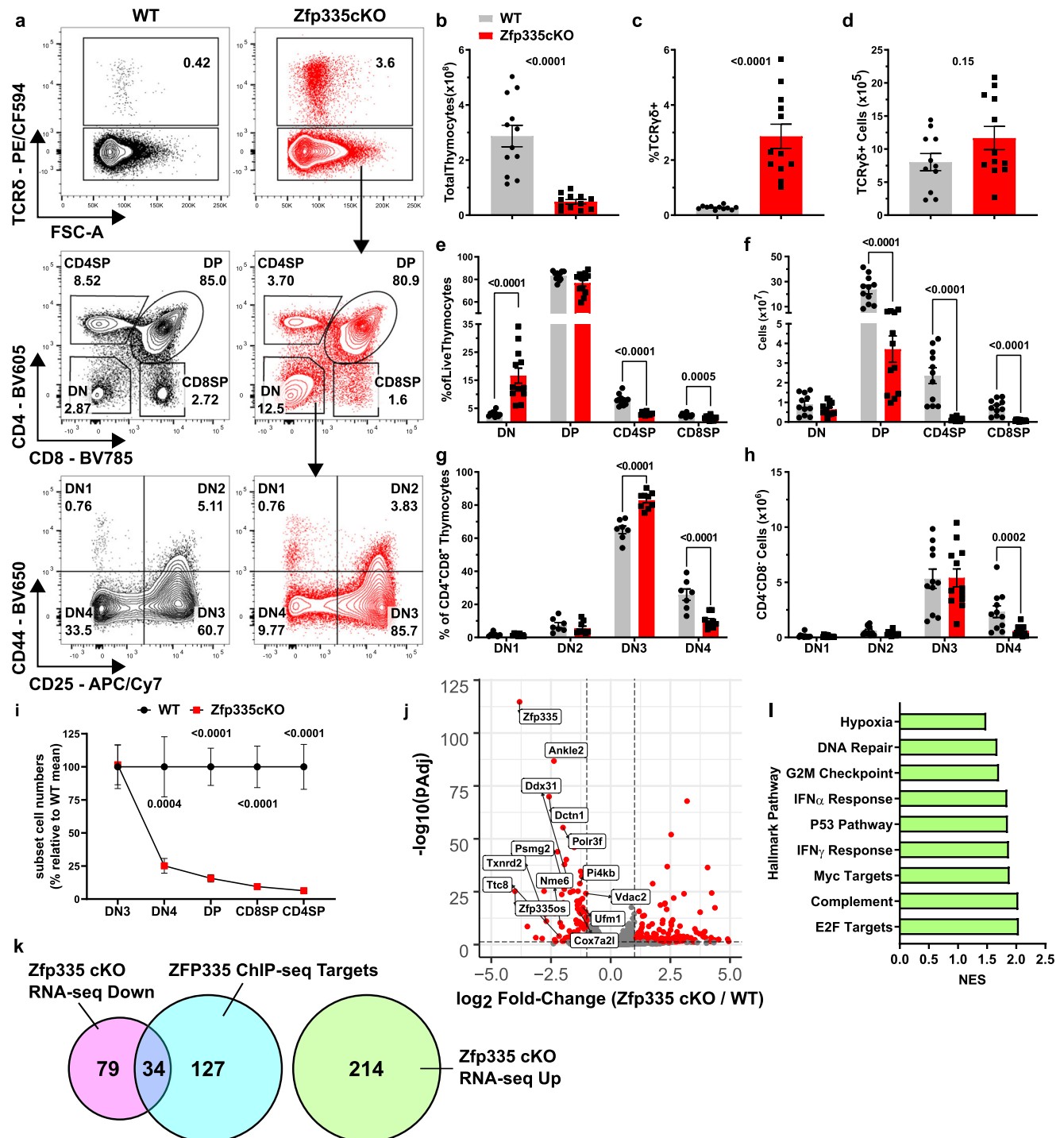

**Fig. 1 | Zfp335 is critical to αβ T-cell development. a** Gating schema for ex vivo analysis thymocyte development beginning with live thymocytes (DAPI⁻ CD90.2⁺, Fig S11a). **b** Total thymic cellularity in WT (Cre-negative) or *Zfp335^fl/fl E8_III cre* (Zfp335cKO) mice. Total numbers (**c**) and frequency (**d**) of TCRγδ⁺ cells in WT or Zfp335cKO thymuses. Numbers (**e**) and frequencies (**f**) of DN, DP, and SP thymocyte subsets in WT or Zfp335cKO thymi. Numbers (**g**) and frequencies (**h**) of early DN1-DN4 thymocyte subsets in WT or Zfp335cKO thymuses. **i** Relative cells numbers in DN3-SP thymocyte subsets represented as percent of WT mean. **j** Volcano plot of differentially expressed genes between Zfp335cKO and WT by RNA-seq. **k** Overlap between Zfp335 ChIP-seq (GSE58293) and differentially expressed genes in

Zfp335cKO and WT DP. **l** Gene Set Enrichment Analysis of differentially expressed genes (**k**). Positive enrichment scores indicate pathways positively enriched in Zfp335cKO cells. **a**–**i** Cre-negative WT (*n* = 11) and Zfp335cKO (*n* = 12) male and female mice from four independent experiments. *P*-values determined by two-tailed Mann–Whitney U-test (**b**–**d**) or two-way ANOVA with post hoc Sidak's test (**e**–**i**). **j**–**l** RNA-seq analysis of *Zfp335^+/+ E8_III cre* or Zfp335cKO DP thymocytes (*n* = 3 each) of 6-week-old female mice from one experiment. Plots show mean ± SEM. Data are compiled from one (**j**–**l**) or 5 independent experiments (**a**–**i**). Source data are provided as a Source Data file.

Zfp335cKO DN3 thymocytes exhibited CD27 upregulation comparable to that of WT (Fig. S3d, e), indicating Zfp335-deficiency does not lead to impaired pre-TCR signaling. Together, these results indicate that the observed reduction of DN4 cells in Zfp335cKO mice did not result from failure to produce TCRβ subunits or failure to transduce pre-TCR signals.

## Zfp335 inhibits apoptosis during the DN-DP transition

Zfp335 deletion during the DN3 stage leads to severe defects in T-cell development, likely during the post-β-selection proliferative phase. To determine if Zfp335-deficiency altered either the proliferation or survival of post-β-selection thymocytes, we directly measured these events in OP9-DL1 cultures in vitro[49]. Consistent with our ex vivo data, Zfp335cKO cells exhibit severely impaired progression to the DP stage (Fig. 2a, b). Zfp335cKO cells exhibited modestly reduced proliferation compared to controls (Fig. 2c, d). In contrast, Zfp335cKO cells underwent substantially increased rates of apoptosis (Fig. 2e, f). Importantly, proliferation tracking (Fig. 2g) and assessment of developmental progression (Fig. 2h) of apoptotic mutant cells demonstrate they have undergone cell division and largely remain DN.

Next, we sought to determine whether loss of Zfp335 also leads to increased rates of cell death among thymocyte populations other than post-β-selection proliferating cells. Zfp335cKO DP cells generated in OP9-DL1 culture exhibited increased rates of apoptosis compared to controls (Fig. S4a). To determine if the same is true in vivo we performed TCRα repertoire analysis using DP thymocytes. Increased rates of Zfp335cKO DP thymocyte apoptosis would yield skewed *Tcra* gene usage towards proximal V and J segments[50]. However, our analyses showed no such skewing (Fig. S1b, c). Furthermore, similar rates of positive selection among DP cells from WT or Zfp335cKO mice were observed (Fig. S1d, e). Together, these data suggest that Zfp335cKO cells are dying during the post-β-selection proliferative phase and that Zfp335 activity promotes the survival of DN4 thymocytes.

## Ectopic Bcl2 expression rescues the developmental defect resulting from loss of Zfp335

Our RNA-seq studies revealed that Zfp335cKO thymocytes exhibit increased expression of the proapoptotic Bcl2-family members PUMA (*Bbc3*), NOXA (*Pmaip1*), and *Bax* (Fig. 3a), suggesting that these factors may be responsible for the observed increase in apoptosis among Zfp335cKO thymocytes. The function of these proteins can be antagonized by the ectopic expression of Bcl2. Thus, we asked whether Bcl2 overexpression could rescue Zfp335cKO thymocyte apoptosis. WT or Zfp335cKO DN3/4 thymocytes were transduced with control or Bcl2-expressing retroviruses and then grown in the OP9-DL1 culture system. Bcl2 overexpression significantly reduced apoptosis in Zfp335cKO cells, indicating the induction of proapoptotic Bcl2-family members was at least partially responsible for the observed increase in apoptosis in Zfp335-deficient thymocytes (Fig. 3b, c).

We next sought to assess the ability of Bcl2 overexpression to rescue Zfp335-deficient cells from apoptosis in vivo through generating Bcl2 conditional transgenic mice (Fig. 3d). Intracellular staining revealed that *Zfp335*[fl/fl] *R26*[LSL-Bcl2-Tg] E8[III]-cre (Zfp335cKO Bcl2-Tg) thymocytes exhibited increased Bcl2 protein expression relative to WT (Fig. 3e). Phenotypic analysis demonstrated that ectopic Bcl2 expression was able to fully rescue the early developmental defects observed in Zfp335-deficient mice, restoring traversal of the β-selection checkpoint, transition to the DP stage, and total thymic cellularity (Fig. 3f[I]L).

Consistent with studies of *Zfp335*[bloto] mice[46], Bcl2 overexpression failed to rescue the impairment in final single positive thymocyte maturation (Fig S5a–c) or peripheral T-cell compartment numbers (Fig S5d, e) and effector status (Fig S5f–h). Taken together, these data suggest that the early impairment of thymocyte development following the loss of Zfp335 expression is due to increased rates of DN4 apoptosis driven by proapoptotic Bcl2-family members. However, our in vivo studies also revealed an additional, Bcl2-independent late block in terminal T-cell differentiation within the thymus.

## Defining the 'true' DN4 thymocyte population at the single-cell level

The DN4 stage of T-cell development remains poorly understood and, as a result, poorly defined. DN4 cells are identified by a lack of expression of identifying markers associated with any other thymocyte subset. Based on these criteria, it is possible that DN4 cells defined by marker exclusion may not be homogenous. To assess whether there is any heterogeneity in the DN4 compartment exacerbated by Zfp335-deficiency, we performed scRNA-seq of phenotypically defined DN4 cells. After quality control, libraries yielded transcriptome data for 6537 or 5392 high-quality cells from WT or Zfp335cKO samples, respectively.

We identified 10 unique cell clusters (Fig. 4a–c). Five clusters were largely cycling cells (DN4_1–5; Fig. 4a, b) uniquely expressing *Ptcra* (pre-Tα) and proliferation-associated genes (*Mki67, Cdk1*) (Fig. 4d), representing bona fide DN4 cells. Three clusters (Mat_1–3) expressed high levels of *Trac and Trbc1* transcripts (Fig. 4d). Two additional clusters (gd17 and gd1) of γδ T cells were identified. gd17 cells express high levels of *Sox13, Rorc,* and *Maf*, features of γδ17, while gd1 express *Nkg7, Il2rb, S1pr1,* and *Il7r* associated with cytotoxic γδ T cells (Fig. 4d). Based on this clustering, Zfp335-deficiency led to substantial proportional increases and decreases in the γδ T-cell clusters and Mat_2 cluster relative to WT control, respectively (Fig. 4c). Consistent with our bulk RNA-seq we found increased IFNα and IFNγ signaling activity in Zfp335cKO Mat and the γδ T-cell clusters (Fig S6).

We were surprised to find a large proportion of phenotypically defined DN4 thymocytes expressing *Trac* transcripts and sought to define these populations. Consistent with their lack of surface CD4 or CD8 these cells uniformly lacked *Cd4, Cd8a,* and *Cd8b1* transcripts (Fig. 4d). We hypothesized that these cells may represent post-positive selection thymocytes that transiently downregulated surface TCR, CD4, and CD8 expression. Consistent with our hypothesis, we found these cells express high levels of *Nr4a1, Cd69, Pdcd1, Egr1, Cd2,* and *Itm2a*, signature genes of positive selection[51] (Fig. 4e). Based on this profile, we define cells from these clusters as maturing αβ T cells. Additionally, we found that Mat_2 cells lack expression of CD24 (Fig. 4e), suggesting that this population may represent mature thymocytes explaining the selective reduction in proportions of these cells in Zfp335cKO samples.

Importantly, most cells associated with the maturing αβ or γδ T-cell clusters were non-cycling (Fig. 4b), and, therefore, not 'true' DN4 cells. Retroviral transduction depends on cell cycling[52]. Therefore, we determined whether 'true' DN4 cells could be separated from contaminating populations ex vivo with retroviruses. Virally transduced or non-transduced DN4 cells were placed in OP9-DL1 culture. Non-transduced DN4 cells preferentially give rise to single-positive cells expressing high levels of surface TCR, whereas transduced DN4 become DP (Fig. 4f, g). Since OP9-DL1 cells are unable to support positive selection, we conclude that these non-transduced DN4 cells are post-positive selection cells transitioning to SP. Together, these results demonstrate that the phenotypically defined DN4 compartment is heterogenous and establishes retroviral transduction as a method to isolate DN4 cells for in vitro analysis.

## Ankle2 is a critical Zfp335-regulated gene required for the survival of DN4 thymocytes

Next, we focused our scRNA-seq analyses on determining the transcriptional changes in DN4 cells resulting from the loss of Zfp335. Maturing αβ and γδ cells were removed, leaving only 'true' DN4 cells. Based on recombination kinetics (Fig S1e), not all Zfp335cKO DN4 cells have undergone deletion. *Zfp335* expression could not reliably delineate mutant from non-mutant cells due to the low detection rate (8% of

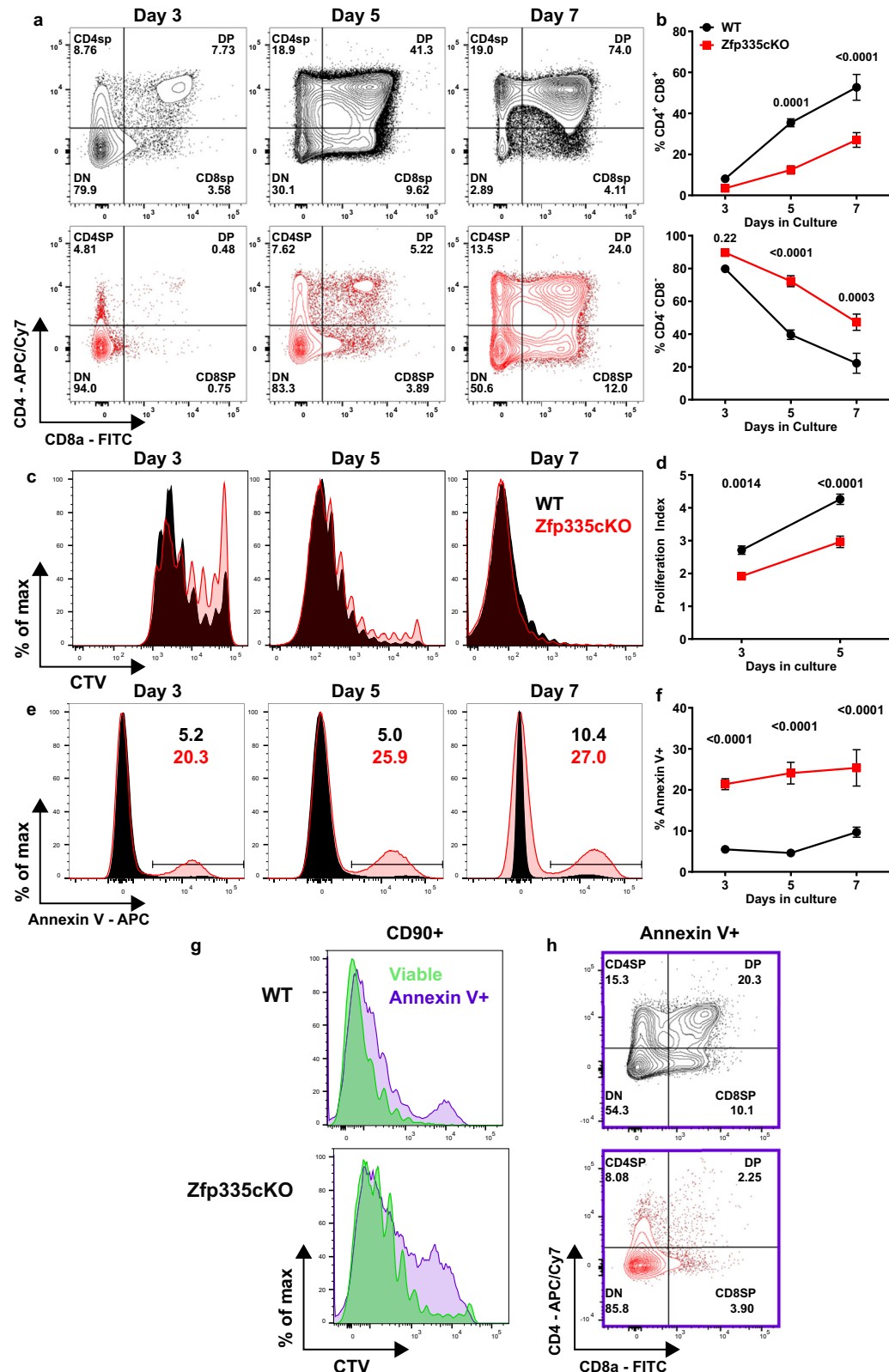

**Fig. 2 | Zfp335cKO DN4 thymocytes undergo increased rates of apoptosis.**
**a, b** Assessment of developmental progression throughout OP9-DL1 culture seeded with WT (black) or Zfp335cKO (red) DN3a thymocytes. Proliferation assessment (**c, d** by Cell Trace Violet (CTV) dilution and apoptosis analysis (**e, f**) based on Annexin V binding at day 3, 5, or 7 of culture. **g** Representative comparison of CTV dilution between Annexin V⁺ and viable (DAPI⁻ Annexin V⁻) cells on day 5 of culture.

**h** Representative CD4 vs CD8 expression among Annexin V⁺ cells on day 5 of culture. $n = 6$ WT or $n = 5$ Zfp335cKO male and female mice from three independent experiments. *P*-values determined using two-way repeated-measures ANOVA with post hoc Sidak's test. Plots show mean ± SEM. Source data are provided as a Source Data file.

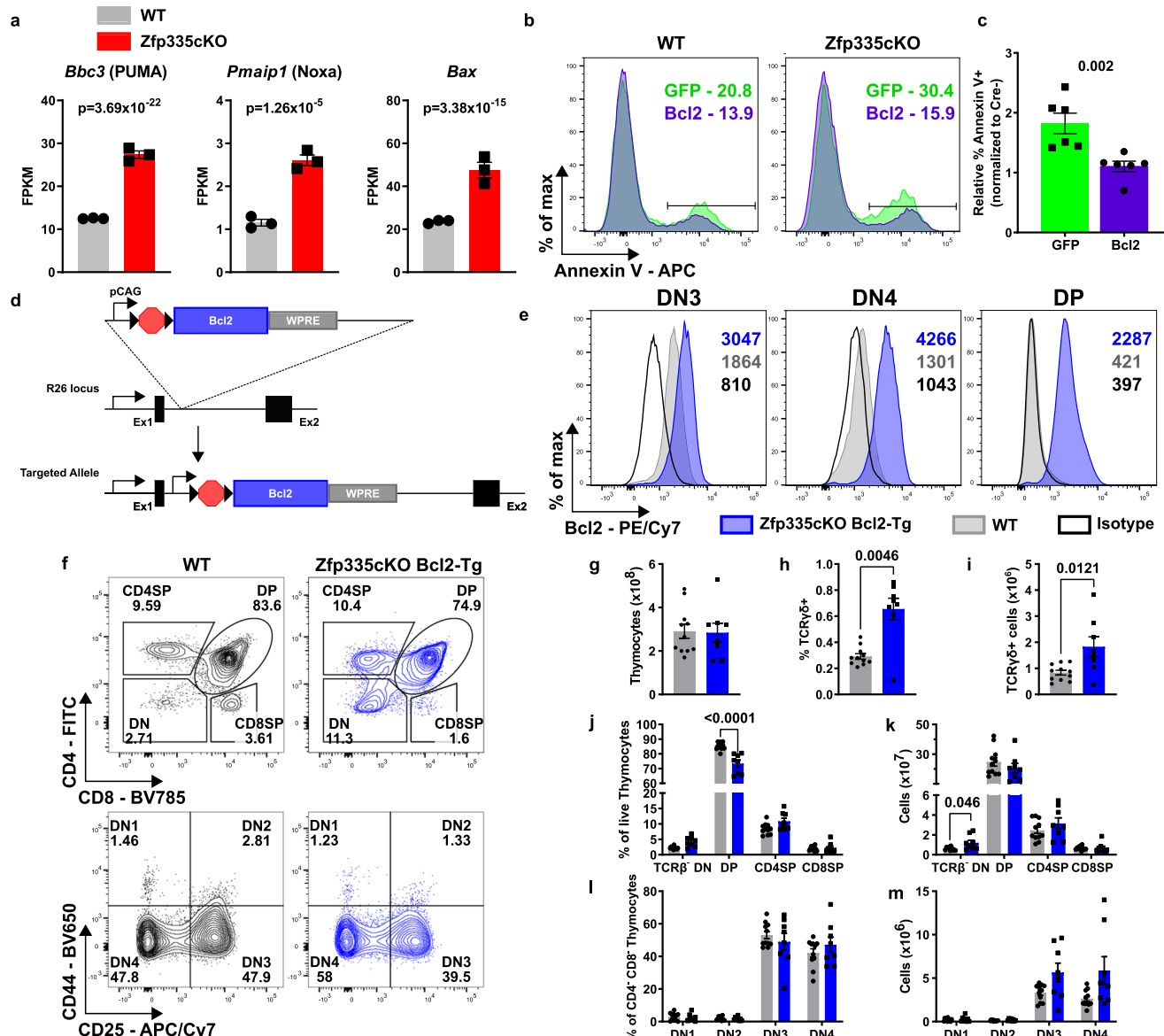

**Fig. 3 | Bcl2 overexpression rescues Zfp335-deficient thymocytes from apoptosis. a** Expression of proapoptotic Bcl2-family genes *Bbc3*, *Pmaip1*, or *Bax* from RNA-seq of control (*n* = 3) or Zfp335cKO (*n* = 3) DP thymocytes. Representative gating (**b**) and quantification (**c**) of apoptosis among Zfp335cKO thymocytes transduced with Bcl2 or GFP RV after 5 days of OP9-DL1 culture (*n* = 5 male and female mice). **d** Schematic diagram of Rosa26^LSL-Bcl2 transgene generation. **e** Representative expression of isotype control (open black) or Bcl2 in WT (gray) or *Zfp335^fl/fl R26^LSL-Bcl2 E8_III-cre* (blue) DN3, DN4, or DP thymocytes. **f** Gating for identification of thymocyte subsets in WT (gray) or *Zfp335^fl/fl R26^LSL-Bcl2 E8_III-cre* (blue) mice.

DN1–4 gating pre-gated on TCRβ⁻. **g** Total thymocyte numbers. Total numbers (**h**) and proportions (**i**) of TCRδ⁺ cells. Frequencies (**j**) and total numbers (**k**) of DN, DP, CD4SP, and CD8SP thymocytes. Frequencies (**l**) and total numbers (**m**) of DN1-DN4 thymocytes. **f–m** *n* = 11 WT or *n* = 8 *Zfp335^fl/fl R26^LSL-Bcl2 E8_III-cre* male and female mice. Data compiled from one (**a**), two (**b**, **c**), or five (**d–l**) independent experiments. *P*-values determined by Wald test with Benjamini–Hochberg correction for multiple testing (**a**), two-tailed Mann–Whitney U-test (**c**), or two-way ANOVA with post hoc Sidak's test (**h–m**). Plots show mean ± SEM. Source data are provided as a Source Data file.

Zfp335cKO vs 17.7% of WT cells). To identify true mutant DN4 cells in our dataset, we assessed transcription factor activity using geneset scores calculated for each cell based on the expression of the Zfp335 ChIP-seq target genes downregulated in mutant DP cells (Fig. 1j, k). Zfp335cKO cells exhibited a bimodal distribution for the geneset. Using established methods[53], cutoff values were determined for the distribution, and cells falling below this threshold were considered true mutants (Fig S7a). Cutoff values were confirmed by differential expression analysis between WT and Zfp335cKO targets high or Zfp335cKO targets low cells. Compared to WT, Zfp335cKO targets low cells exhibited differential expression of 80 genes (60 down, 20 up; Fig S7b), whereas Zfp335cKO targets high cells only exhibited differential expression of 7 genes (5 down, 2 up; Fig S7c).

Zfp335cKO cells above the threshold were considered non-mutant, removed, and the remaining cells were then reanalyzed, identifying eight unique clusters (Fig S7d). WT and mutant cells were distributed across each cluster. C1–3 were enriched for WT, whereas C4 was almost entirely mutant cells (Fig S7e). Despite regression of standard cell-cycle-associated genes, clustering was largely dictated by cell cycle (Fig S7f–i). We observed no differences in cell cycle phase distributions between WT and mutant (Fig S7h). Therefore, we chose to compare WT and mutant DN4 cells based on genotype (Fig. 5a). Among the 60 downregulated genes in mutant DN4 cells, 44 are Zfp335 targets by ChIP-seq (Fig. 5b)[46]. We hypothesized that reduced expression of one or more of these genes was responsible for the increased rates of apoptosis observed in mutant DN4 cells. Thus, we

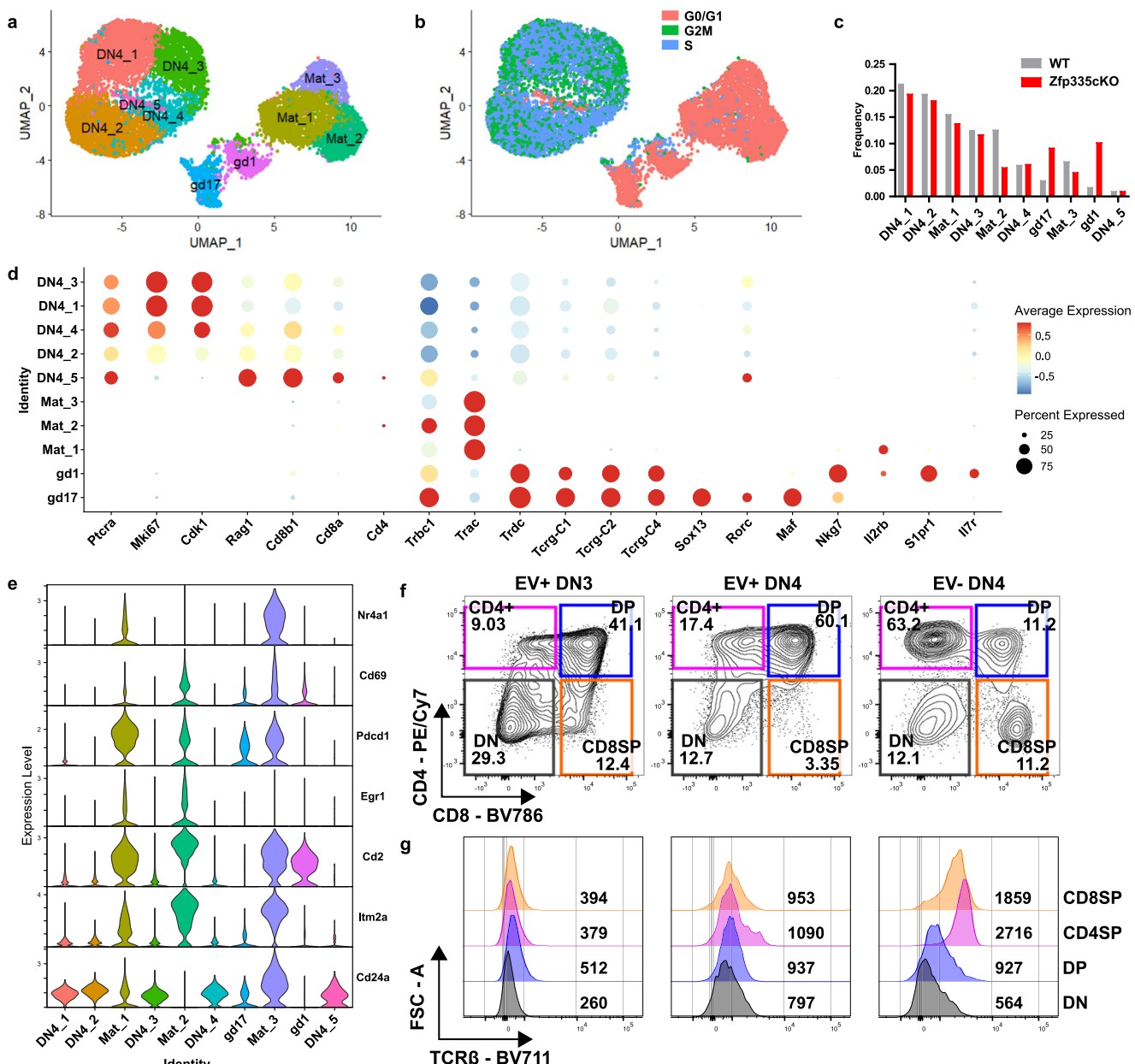

**Fig. 4 | Defining the 'true' DN4 thymocyte population at the single-cell level.**
**a** UMAP projection and identification of 10 clusters identified in full scRNA-seq dataset. **b** UMAP colored by cell cycle phase. Blue or green identify actively cycling cells. **c** Frequency distributions for WT ($n = 6357$) and Zfp335cKO ($n = 5392$) cells across the ten clusters. **d** Dot plot of key cell-type-defining genes. **e** Violin plots of positive selection signature genes in thymocytes[51]. **f** Representative gating for CD4 vs. CD8 expression on day 3 of OP9-DL1 cultures seeded with WT Thy1.1 retrovirus transduced (EV+) DN3 or DN4 cells or non-transduced (EV−) DN4 cells. **g** Representative TCRβ expression among DN, DP, CD4SP, or CD8SP cells from **f**. Numbers indicate geometric MFI of TCRβ expression. **a–e** Data are from one experiment. **f, g** Data representative of two independent experiments. Source data are provided as a Source Data file.

examined the expression of the 12 Zfp335 target genes with experimental evidence demonstrating a negative regulatory role in cell death (Fig. 5c, d). Four exhibited reduced expression in mutant DN4 thymocytes (Fig. 5c). Examination of expression frequency identified *Ankle2* as having the greatest reduction in percent of mutant cells expression (Fig. 5e). Interestingly, pathway analysis did not show any enrichment for apoptosis among Zfp335cKO mutant cells (Fig S7j).

*Ankle2* encodes an ER-restricted ankyrin repeat and LEM domain-containing protein[54]. *Ankle2* was recently identified as a critical Zfp335-regulated factor in the establishment of the naïve T cell[46]. Therefore, we tested whether *Ankle2* overexpression could rescue Zfp335cKO apoptosis. WT or Zfp335cKO DN3 thymocytes were transduced with EV or Ankle2 retrovirus and cultured on OP9-DL1 cells. Importantly, *Ankle2* overexpression was able to fully rescue Zfp335-deficient

thymocytes from increased rates of apoptosis (Fig. 5f, g). Moreover, *Ankle2* overexpression led to significantly increased proportions of DP cells among Zfp335cKO samples (Fig. 5h, Fig S8a).

Next, we sought to confirm that Ankle2 expression is directly regulated by Zfp335 in pre-T cells. Analysis of published ChIP-seq data showed Zfp335 binds the proximal promoter of *Ankle2* in thymocytes (Fig. 5i). Similar to *Zfp335*, *Ankle2* is ubiquitously expressed throughout T-cell development (Fig S8b). To examine the relationship between *Zfp335* and *Ankle2* expression, we utilize the DN4-like mouse thymocyte cell line *Scid.adh.2c2*[55] for CRISPR-based transcriptional inhibition (CRISPRi) studies[56]. These cells were transduced with retroviruses expressing *Zfp335* promoter-targeting gRNA and anti-GCN4scFv-sfGFP-KRAB fusion construct. *Zfp335*-targeted cells exhibited reduced *Ankle2* expression proportional to the

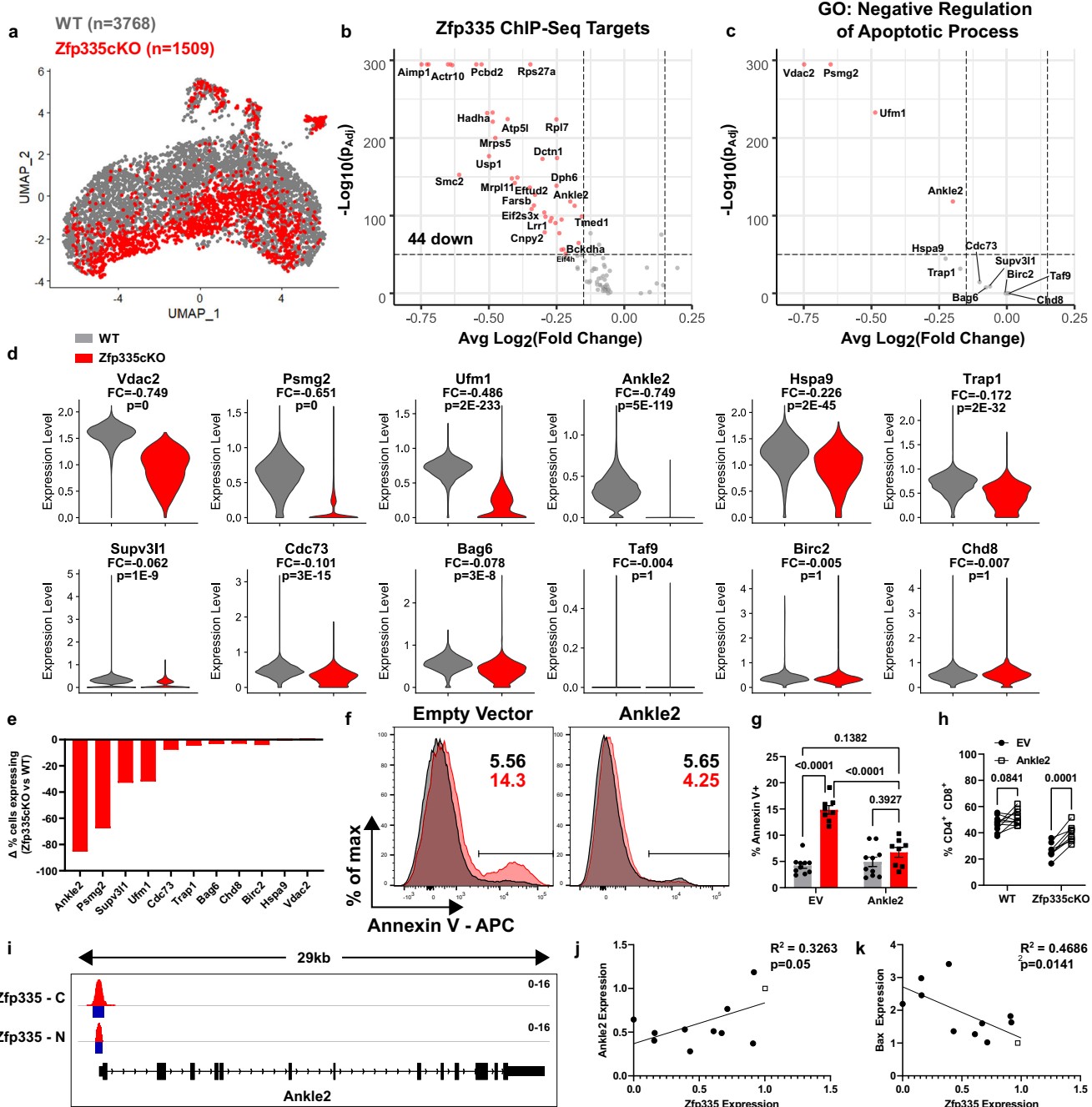

**Fig. 5 | scRNA-seq identifies Ankle2 as a critical Zfp335-regulated gene controlling survival of DN4 thymocytes. a** UMAP projection of WT and 'true' Zfp335 mutant DN4 cells colored by genotype. Volcano plot of all differentially expressed Zfp335 target genes (**b**) or those experimentally shown negatively regulate apoptotic processes (**c**) between Zfp335 mutant and WT cells. **d** Violin plots of antiapoptotic Zfp335 target gene expression between Zfp335 mutant and WT DN4 cells. FC indicates $\log_2$(Fold Change) between Zfp335cKO and WT, p indicates adjusted *p*-values. **e** Differential proportions of Zfp335 mutant cells expressing antiapoptotic genes from **c**, **d** compared to WT cells based on dropout-imputation. Representative gating (**f**) and quantification of apoptosis (**g**) or DP cell frequency (**h**) for EV or Ankle2 retrovirus transduced WT (*n* = 10) or Zfp335cKO (*n* = 8) DN3

thymocytes from male and female mice cultured on OP9-DL1 cells for 3 days. **i** Zfp335 ChIP-seq track of *Ankle2* locus in WT thymocytes (Zfp335-C or Zfp335-N antibodies, GSE58293). Blue boxes indicate significant binding peaks. Correlation between *Ankle2* (**j**) or *Bax* (**k**) and *Zfp335* expression in Scid.adh.2c2.SunTag CRIS-PRi cells expressing nontargeting (open squares, *n* = 2) or Zfp335-targeting (closed circles, *n* = 10) gRNAs. Data are compiled from one (**a**–**e**), two (**j**, **k**), or three (**f**–**h**) independent experiments. *P*-values determined by two-sided Wilcoxon Rank Sum test (**b**–**d**), two-way ANOVA with post hoc Tukey's test for multiple comparisons (**g**), repeated-measures ANOVA with post hoc Sidak's test (**h**), or simple linear regression (**j**, **k**). Plots show mean ± SEM. Source data are provided as a Source Data file.

efficiency of *Zfp335* knockdown (KD) (Fig. 5j). Additionally, *Zfp335*KD resulted in increased expression of *Bax* like that observed in Zfp335cKO thymocytes (Fig. 5k). Together, these results demonstrate a direct relationship between *Zfp335* and *Ankle2* expression in developing T cells and suggest reduced *Ankle2* expression resulting from loss of Zfp335 drives DN4 apoptosis in Zfp335cKO mice.

## Loss of Zfp335 disrupts nuclear envelope architecture and promotes cGAS/STING signaling

Next, we sought to determine the mechanism driving this increase in cell death resulting from reduced *Ankle2* expression. Ankle2 has previously been shown to control nuclear envelope (NE) reassembly and integrity following mitosis through regulation of Barrier to

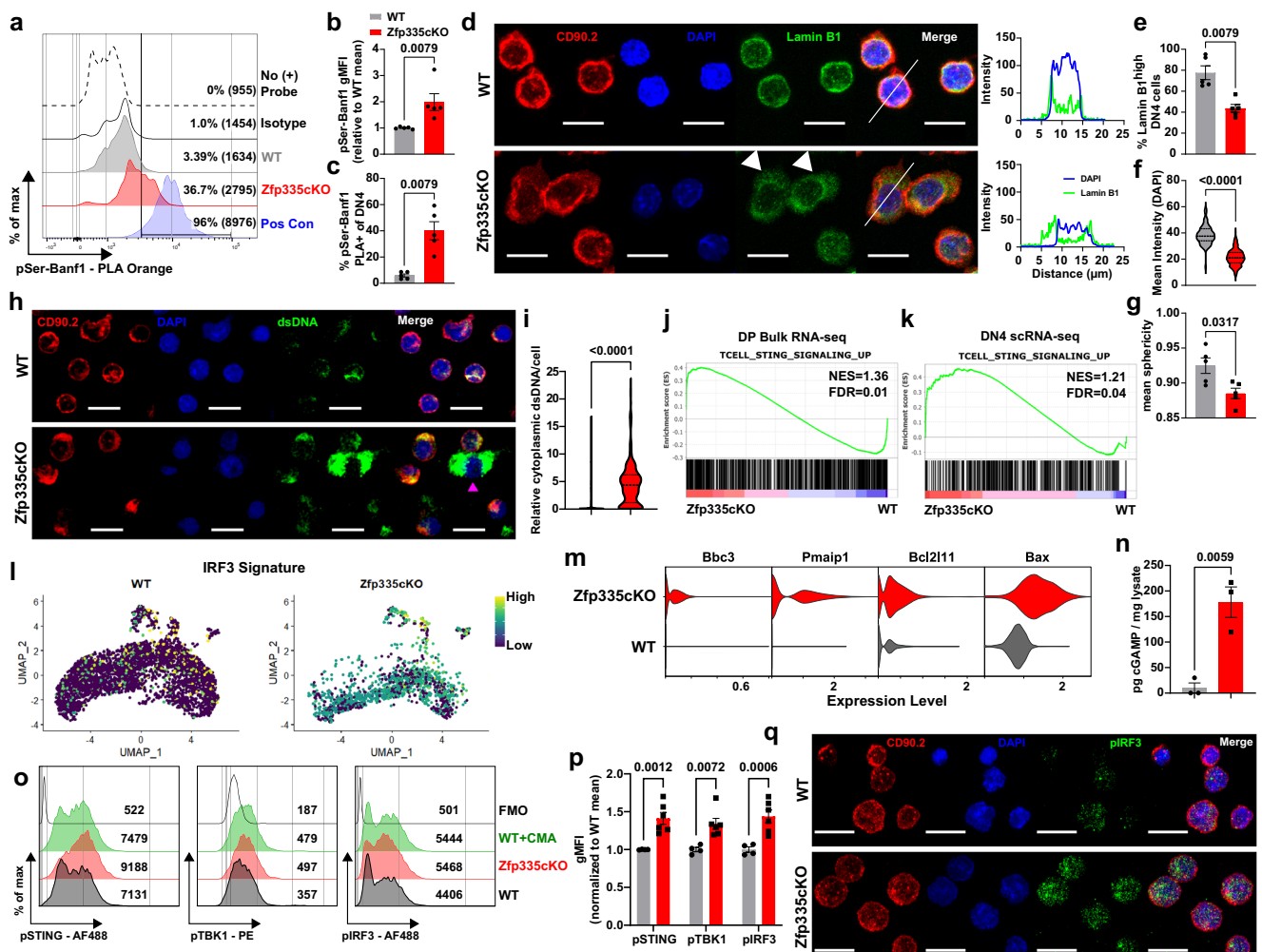

**Fig. 6 | Loss of Zfp335 leads to altered nuclear envelope architecture, accumulation of cytosolic dsDNA and promotes cGAS/STING signaling.**
**a** Representative histograms and gating of Baf phosphorylation as measured by proximity ligation assay (PLA). Percent phosphoserine-Baf and geometric MFI in parenthises are shown. Phosphoserine-Lamin B1 PLA was used as positive control. Quantification of Baf phosphorylation based on geometric MFI (**b**) or percent positive cells (**c**). $n = 5$ mice per genotype. **d** Representative immunofluorescence images of full cell thickness maximum intensity projections (left) and profile plots (right) of nuclear envelope staining in ex vivo DN4 thymocytes. Profile plots are based on white lines shown in merged images. Scale bars represent 10 μm. Quantification of frequency of cells with high nuclear-associated Lamin B1 (**e**), mean DAPI pixel intensity (**f**) or mean nucleus sphericity (**g**) for ex vivo DN4 thymocytes. **e, g** $n = 5$ mice per genotype, **f** $n = 124$ WT and $n = 490$ Zfp335cKO cells. Representative images (**h**) and quantification (**i**) of cytoplasmic dsDNA in WT ($n = 548$ cells, 4 mice) or Zfp335cKO ($n = 268$ cells, 6 mice) thymocytes following 3 days in OP9-DL1 culture. Magenta arrow indicates OP9-DL1 cell. GSEA enrichment plots for T-cell-specific STING signaling gene signature in DP bulk (**j**) or DN4 scRNA-seq datasets (**k**). **l** UMAP projection of IRF3 gene signature in WT or Zfp335 mutant DN4 thymocytes. **m** Violin plots of proapoptotic Bcl2 gene expression in WT or Zfp335 mutant DN4 thymocytes. **n** Normalized cGAMP concentration for WT ($n = 3$ mice) or Zfp335cKO ($n = 3$ mice) thymocytes. Representative histograms (**o**) and quantification (**p**) of phospho-STING, -TBK1, and -IRF3 in WT ($n = 4$ mice) or Zfp335cKO ($n = 6$ mice) thymocytes following 3 days in OP9-DL1 culture. WT thymocytes treated with 250 μg/mL cridanimod (WT + CMA) for 2 h were used as a positive control. **q** Representative images of phospho-IRF3 staining related to o,p. *P*-values determined by two-tailed Mann–Whitney U-test (**b–i**), unpaired two-tailed Student's *t*-test (**n**), or two-way ANOVA with post hoc Tukey's test (**m**) or Sidak's test (**p**). Data shown are compiled from one (**j–m**), two (**h**, **i**, **n–p**), or three (**a–g**) independent experiments utilizing male and female mice. Plots show mean ± SEM (**b, c, e–g, n, p**) or median and interquartile range (**f, i**). Source data are provided as a Source Data file.

---

Autointegration Factor 1 (Banf1 or Baf) phosphorylation[54]. Consistent with reduced *Ankle2* expression, we observed significant increases in Baf phosphorylation among Zfp335cKO DN4 thymocytes (Fig. 6a–c). Additionally, as previously reported[57], disruption of *ANKLE2* or *BANF1* expression in Hela cells led to severe disruptions in NE architecture (Fig S8b). To determine if the same is true for *Zfp335*-deficient DN4 thymocytes, we examined the NE ex vivo. Indeed, Zfp335cKO DN4 thymocytes exhibit significantly altered NE architecture characterized by diffuse Lamin B1 throughout the cytosol, reduced DAPI signal possibly the result of loss of nucleocytosoplasmic compartmentalization and reduced nuclear sphericity (Fig. 6g). To test whether the observed NE defects result in loss of nucleocytoplasmic compartmentalization we measured the abundance of cytosolic dsDNA in WT or Zfp335cKO

thymocytes by confocal microscopy. Consistent with altered NE architecture, we observed a significantly increased abundance of cytoplasmic dsDNA in Zfp335cKO compared to WT thymocytes (Fig. 6h, i). Together, these data confirm that loss of Zfp335 leads to significantly altered NE architecture and accumulation of cytoplasmic dsDNA consistent with dysregulation of Ankle2/Baf-mediated NE reassembly and maintenance.

Accumulation of cytosolic DNA or exposure of nuclear contents to the cytosol via NE disruption has been shown to activate the cGAS/STING pathway[58,59]. In T cells, cGAS/STING signaling generally results in anti-proliferative and proapoptotic effects[35,36,38,60]. Therefore, we hypothesized that NE defects resulting from disruption of the Ankle2-Banf1 pathway downstream of Zfp335 loss drive cGAS/STING

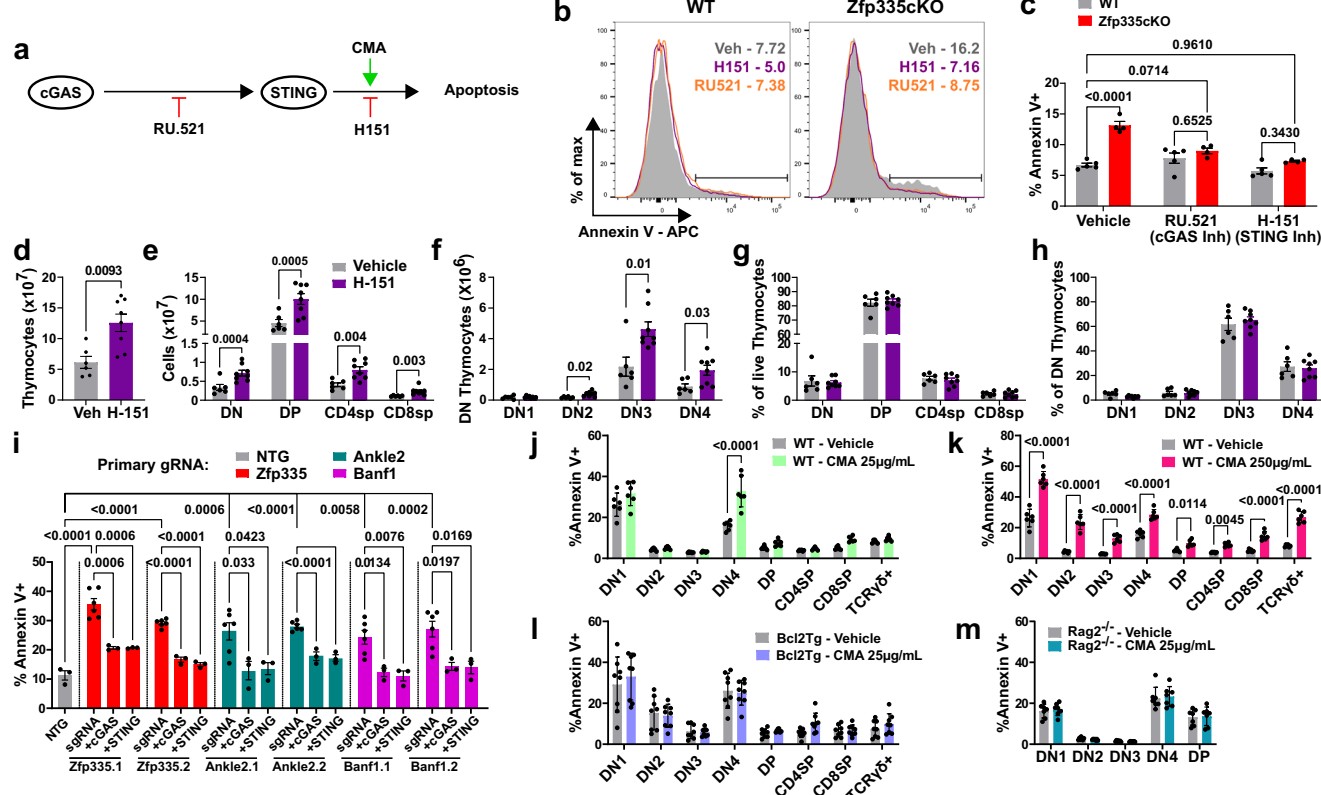

**Fig. 7 | The Zfp335/Ankle2/Baf axis suppresses cGAS/STING-mediated apoptosis of DN4 thymocytes. a** Schematic diagram of inhibitors (RU.521 or H-151) or agonists (CMA) used to study cGAS/STING-dependent apoptosis of DN4 thymocytes. Representative histograms (**b**) and quantification (**c**) of Annexin V binding for WT (*n* = 5 mice) or Zfp335cKO (*n* = 4 mice) DN4 thymoctes treated with cGAS (RU.521) or STING (H-151) inhibitors or vehicle control and cultured on OP9-DL1 stromal cells for 3 days. Total thymocyte (**d**), DN, DP, CD4SP, and CD8SP or DN1-DN4 cell numbers (**e, f**) or frequencies (**g, h**) for Zfp335cKO mice treated with H-151 (*n* = 8 mice) or vehicle (*n* = 6 mice) in vivo for 7 days. **i** Quantification of Annexin V binding among DN4 cells from *R26^{LSL-Cas9} Tcrd^{CreERT2}* thymocytes transduced with gRNA-expressing retroviruses and cultured for three days on OP9-DL1 cells with

4-hydroxytamoxifen. *n* = 3 biological replicates pooled from 3 mice each for each dual gRNA transduction. Percent apoptosis induced by small molecule activation of STING among WT thymocyte subsets treated with 25 μg/mL (**j**) or 250 μg/mL CMA (**k**), and Zfp335cKO Bcl2-Tg (**l**), or αCD3-treated Rag2^{−/−} thymocyte subsets (**m**) treated with 25 μg/mL CMA. Panels **j** and **k** were from the same experiments. *n* = 6 mice (**j, k**), *n* = 8 mice (**l**), or *n* = 7 mice (**m**). *P*-values determined by two-tailed Mann−Whitney U-test (**d**) or two-way ANOVA with post hoc Tukey's test (**c**) or Sidak's test (**e**–**h, j**–**m**) or one-way ANOVA with post hoc Tukey's test (**i**). Data shown are compiled from two (**b, c, m**), three (**i**–**l**), or five (**d**–**h**) independent experiments utilizing male and female mice. Plots show mean ± SEM. Source data are provided as a Source Data file.

activation. Consistent with this hypothesis, GSEA revealed enrichment for genes upregulated by T cells in response to STING signaling in both our bulk DP and single-cell DN4 datasets (Fig. 6j, k). Additionally, we found increased IRF3 activity among mutant cells (Fig. 6l). cGAS/STING-mediated death of mature T cells occurs in part due to increased expression of proapoptotic Bcl2-family genes[36]. Like our findings from bulk RNA-seq (Fig. 3a), we also observed increased expression of *Bbc3* (PUMA), *Pmaip1* (NOXA), *Bcl2l11* (Bid), and *Bax* among Zfp335cKO DN4 cells in our scRNA-seq dataset (Fig. 6m). In agreement with our bioinformatic analyses, we observed significantly increased cGAMP levels (Fig. 6n), phosphorylation of STING, TBK1 and IRF3 in Zfp335cKO thymocytes (Fig. 6o, p) as well as nuclear translocation of phosphorylated IRF3 (Fig. 6q).

In addition to nuclear DNA, mitochondrial DNA (mtDNA) serves as a substrate for cGAS[61]. mtDNA release requires mitochondrial outer membrane permeabilization resulting in mitochondrial membrane depolarization[62]. Examination of mitochondria showed Zfp335cKO thymocytes exhibit normal mitochondrial membrane potential and total mitochondrial mass (Fig. S8d, e). Therefore, mtDNA release is unlikely to be driving cGAS/STING-mediated death following the loss of Zfp335. Together, these data suggest that exposure of gDNA to cytosolic cGAS resulting from disrupted nuclear envelope architecture is the most likely cause of cGAS/STING signaling resulting from Zfp335 deletion.

## Disruption of the Zfp335/Ankle2/Baf axis drives cGAS/STING-dependent apoptosis of DN4 thymocytes

Next, we sought to determine the functional importance of enhanced cGAS/STING signaling in T-cell development following the loss of Zfp335. To test this, 'true' DN4 cells were isolated by EV viral transduction and then placed in OP9-DL1 culture for 3 days with small molecule inhibitors of cGAS (RU.521)[63] or STING (H-151)[64]. Chemical inhibition of either cGAS or STING fully rescued Zfp335cKO DN4 cells from death (Fig. 7a–c). Additionally, Zfp335cKO mice receiving H-151 for 7 days exhibited significantly increased numbers of total thymocytes compared to vehicle controls (Fig. 7d). Importantly, this increase in cellularity was primarily due to increased DP numbers (Fig. 7e–h, Fig. S10a). Due to the short duration of treatment, we conclude that the increase in DP cells among H-151-treated Zfp335cKO mice is the result of reduced cell death during the preceding proliferative DN4 stage.

cGAS/STING signaling is known to drive type-I interferon (IFN-I) responses[65] and mTOR activation[66]. While we did not observe altered transcriptional signatures associated with mTOR activity or IFN-I signaling in our DN4 scRNA-seq dataset (Fig. S9a, b), the IFN-I signaling program was significantly enriched among Zfp335cKO DP thymocytes compared to control (Fig. 1l). To assess the importance of these pathways in Zfp335cKO cell death we performed OP9-DL1 cultures in the presence of IFNAR-1 blocking antibody or mTOR inhibiting small

molecules. Inhibition of either pathway had no impact on rates of Zfp335cKO cell death (Fig. S9c−e). Together these data support the conclusion that cGAS/STING signaling resulting from loss of Zfp335 drives apoptosis of DN4 thymocytes independent of IFN-I and mTOR signaling. Instead, cell death is likely driven by the induction of proapoptotic members of the Bcl2 family.

Next, we sought to determine the role of the Zfp335/Ankle2/Baf axis in suppressing the cGAS/STING-mediated apoptosis in DN4 cells. To test this, $R26^{LSL-Cas9}$ $Tcrd^{CreERT2}$ DN3/DN4 thymocytes[67] were transduced with retroviruses expressing *Zfp335, Ankle2, or Banf1* (encoding Baf) and *Mb21d1* (encoding cGAS) or *Tmem173* (encoding STING)-targeting gRNAs or nontargeting control gRNAs (NTG) then cultured for three days with OP9-DL1 cells in the presence of 4-hydroxytamoxifen. Consistent with conditional deletion, Cas9 targeting of *Zfp335* led to a substantial increase in DN4 apoptosis (Fig. 7i, Fig S10b). Additionally, targeting of *Ankle2* or *Banf1* similarly lead to increased DN4 apoptosis. Importantly, these increases in apoptosis were cGAS/STING-dependent (Fig. 7i, Fig S10b). Similar results were observed when Cas9 expression was controlled by E8III-cre (Fig S10c, d). Together, these results demonstrate that disruption of the Zfp335/Ankle2/Baf axis drives cGAS/STING-mediated apoptosis of post-β-selection DN4 thymocytes.

### DN4 thymocytes exhibit increased sensitivity to cGAS/STING-mediated cell death

Finally, we sought to determine whether sensitivity to cGAS/STING-driven cell death is a unique feature of Zfp335cKO DN4 cells or a mechanism of the DN4 stage. DN-enriched WT thymocytes were treated with the STING agonist cridanimod (CMA) overnight and then assayed for apoptosis. Interestingly, we found DN4 cells are uniquely sensitive to STING-mediated apoptosis at low concentrations of STING agonist (Fig. 7j, Fig S11a). However, increasing the STING agonist concentration 10-fold was sufficient to promote apoptosis of all thymocyte subsets (Fig. 7k, Fig S11a). Additionally, the viability of Zfp335cKO Bcl2-Tg thymocytes was not impacted by CMA treatment (Fig. 7l, Fig S11b), suggesting that induction of proapoptotic Bcl2-family members downstream of STING activation is necessary for apoptosis of DN4 thymocytes.

We were surprised at the unique sensitivity of DN4 cells to cGAS/STING-mediated apoptosis. We hypothesized that such sensitivity may be due to the presence of TCR excision circles (TRECs) generated by V(D)J recombination at the preceding DN3 stage, which may be exposed to the cytosol during cell division, thereby reducing the signaling threshold for STING-mediated cell death. To test this, we generated post-β-selection thymocytes from $Rag2^{-/-}$ mice by intraperitoneal injection of αCD3ε antibody. Following antibody treatment, thymocytes were harvested, subjected to STING agonist treatment ex vivo, and assessed for apoptosis. Consistent with our hypothesis, Rag-deficient DN4 cells do not exhibit increased sensitivity to STING-mediated cell death compared to vehicle control (Fig. 7m, Fig S11c). Together, these data demonstrate that activation of the cGAS/STING pathway is a major contributor to Zfp335cKO DN4 apoptosis and that WT DN4 cells exhibit increased sensitivity to cGAS/STING-mediated death, possibly a result of V(D)J recombination at the DN3 stage.

Altogether, our studies reveal that loss of Zfp335 leads to defective T-cell development resulting from dysregulation of the Zfp335/Ankle2/Baf axis, ultimately driving cGAS/STING-mediated DN4 cell death.

## Discussion

In this study, we identify Zfp335 as a critical transcription factor regulating early T-cell development within the thymus. Specifically, it functions to promote the survival of proliferating cells following β-selection. Conditional deletion of Zfp335 led to severe reductions in all T-cell populations beginning at the DN4 stage of development.

Mechanistically, we show that reduced expression of the Zfp335-regulated gene *Ankle2* is responsible for increased sensitivity to cell death and disruption of the Zfp335/Ankle2/Baf pathway controlling NE architecture drives cGAS/STING-dependent DN4 apoptosis.

Our studies highlight the necessity of sustained Zfp335 and Ankle2 expression to promote DN4 survival and support proper T-cell development. Zfp335 is ubiquitously expressed throughout T-cell development. Therefore, it is likely that Zfp335 plays numerous roles throughout thymopoiesis. In addition to supporting DN4 survival, we show that Zfp335 is also required for terminal thymocyte maturation. Based on the mechanism uncovered in our work and the high degree of functional similarity between DN4 and DN2 cells, it is likely that loss of Zfp335 at the DN1 stage or earlier may block T-cell development at the DN2 stage. Such a hypothesis is supported by a recent study that found that mice carrying a human disease associated STING mutation (V154M) exhibit severe T lymphopenia due to increased rates of early thymic progenitor apoptosis[68]. However, to test this hypothesis, alternative means of Zfp335 deletion will need to be utilized.

Our studies provide a comprehensive assessment of the heterogeneity within the DN4 thymocyte compartment at the single-cell level. Surprisingly, phenotypically defined DN4 cells consist of cycling cells expressing pre-Tα which represent 'true' DN4 cells and mature or maturing αβ and γδ T cells. Positive selection of DP thymocytes induces a slight, transient downregulation of CD4 and CD8[69], however, the maturing αβ cells identified in our dataset completely lack both protein and mRNA expression. The cells we identified expressing TCRα transcripts exhibited expression patterns consistent with positive selection[51] and, therefore, are likely post-positive-selection cells that have transiently lost surface expression of TCR, CD4, and CD8. Alternatively, these cells may have undergone positive selection without ever expressing CD4 or CD8. Regardless, these maturing cells may represent a key developmental path within the thymus. However, more detailed studies will be needed to fully characterize these cells and determine if they represent a unique lineage or simply a rare differentiation path that can be taken by any positively selected cell.

Han et al. recently identified a hypomorph allele of *Zfp335* (*Zfp335bloto*) as the causative mutation leading to reduced total peripheral T cells and an almost complete absence of naïve T cells[46]. They found *Ankle2* to be a critical Zfp335-regulated gene controlling the late stages of thymic T-cell maturation. However, the mechanism by which Ankle2 regulates maturation, and the establishment of the naïve T-cell compartment remains unclear. The lack of apparent developmental defects in *Zfp335blt/blt* mice during early T-cell development is likely due to their use of a hypomorph allele instead of a conditional knockout, as *Zfp335blt/blt* mice exhibited normal expression of Ankle2 during the DN4 stage.

We have shown that *Zfp335* expression is, at least partially, regulated by E-protein activity in developing T cells. E proteins play numerous indispensable roles throughout organismal development, including T-cell development[4,27,39–42,70–72]. However, due to widespread binding throughout the genome, the roles of transcriptional networks established by E proteins remain incompletely understood[42]. Our studies identify Zfp335 as a key transcription factor downstream of E proteins critical to T-cell development.

To date, studies of T-cell-intrinsic roles for cGAS/STING pathway have largely focused on activation via synthetic STING agonists[36,38,60] or expression of constitutive gain-of-function STING mutations[35]. These studies have primarily focused on the roles of this pathway in mature peripheral T cells. To our knowledge, this may be the first report of a physiological role for cGAS/STING in T-cell development. Additionally, our identification of the Zfp335/Ankle2/Baf axis as key in the repression of cGAS reveals that transcriptional pathways can prevent cGAS activation by self-DNA independent of regulating cGAS or STING expression.

Baf was recently identified as a key inhibitor of cGAS sensing of self-DNA through competitive binding[59]. The ability of Baf to bind DNA is dependent upon its dephosphorylation, which has been shown to be controlled by Ankle2 during mitotic exit[54]. Therefore, we propose the following mechanism by which loss of Zfp335 drives cGAS/STING-mediated apoptosis of DN4 thymocytes. Loss of Zfp335 results in impaired *Ankle2* expression, which in turn leads to the failure of Baf dephosphorylation during cell division. Baf hyperphosphorylation leads to improper NE reassembly and can drive spontaneous NE rupture exposing nuclear DNA to the cytosol allowing unrestricted cGAS activation and STING-mediated apoptosis.

A surprising finding of our study was the unique sensitivity of DN4 thymocytes to cGAS/STING-mediated apoptosis. We did not observe significant increases in cell death among any other population of thymocytes following low-dose ex vivo STING activation. However, 10-fold higher STING agonist concentrations led to significant increases in cell death across all thymocyte subsets. This sensitivity was efficiently abrogated upon Bcl2 overexpression, suggesting a dominant role of proapoptotic members of this protein family in driving cell death. Interestingly, we found that preventing V(D)J recombination via deletion of *Rag2* was sufficient to protect DN4 cells from STING-mediated cell death, suggesting that V(D)J recombination may play a role in this unique sensitivity to cGAS/STING signaling. A reasonable explanation for our finding is that TRECs generated by V(D)J recombination may become localized to the cytoplasm upon cell division at the DN4 stage, thereby providing activating signals for cGAS. This cGAS activity, in turn, may lower the signaling threshold for STING-mediated cell death. However, this does not explain the lack of DP cell sensitivity to STING agonists, which undergo repeated rounds of VJ recombination. We propose that the apparent lack of sensitivity to STING signaling among DP cells is due to the loss of cGAS and STING expression during the DN-DP transition[51]. Should this be true, it is reasonable to hypothesize that the downregulation of cGAS and STING expression is necessary to facilitate the survival of DP thymocytes. Alternatively, the lack of cell division during T-cell development beginning upon initiation of TCRα recombination may prevent exposure of DP-derived TRECs to the cytosol. In either case, further study of this phenomena is warranted.

Consistent with studies of cGAS/STING signaling in mature T cells[35,38], we found cGAS/STING-mediated apoptosis in DN4 cells is independent of IFN-I signaling. Interestingly, we found that Zfp335-deficient DP and DN4-like maturing T cells exhibit increased IFN-I signaling activity compared to controls. Similar transcriptional activity was previously observed in *Zfp335*[bloto] mice[46]. While tonic IFN-I signaling is critical to normal T-cell development[73], enhanced signaling has been shown to severely impair thymopoiesis[74]. The mechanism by which Zfp335 and Ankle2 regulate terminal T-cell maturation in the thymus is unclear. However, it is possible that enhanced IFN-I signaling resulting from cGAS/STING activation may contribute to this defect.

Interestingly, in humans, ANKLE2 is a target of Zika virus protein NS4A, which antagonizes its activity, ultimately leading to microcephaly[75]. Humans carrying homozygous or compound heterozygous mutations in either *ZNF335* or *ANKLE2* exhibit severe microcephaly like that characteristic of Zika patients[43,76]. Recent studies have demonstrated a critical role for central nervous system immune cells in regulating neuronal stem cell maintenance and differentiation. Specifically, microglia play a key role in this process[77–79]. Under conditions that stimulate cGAS activity, microglia and other CNS immune cells preferentially undergo apoptosis[80]. Based on the mechanism revealed in this study, it is possible that microcephaly resulting from Zika infection or loss of ZNF335 or ANKLE2 may be driven by cGAS/STING-dependent apoptosis of neuronal progenitors and/ or CNS immune cells. Should our mechanism extend to neuronal progenitors or CNS immune cells, it may be possible to pharmacologically prevent microcephaly in these specific instances by inhibition of the cGAS/STING pathway. However, further research will be required to determine the viability of such a therapeutic approach.

Beyond CNS development, the mechanism uncovered through our studies may apply to more diverse biological phenomena. Mutations in BAF have previously been shown to drive Nestor-Guillermo Progeria Syndrome, a disease associated with severe premature aging which typically manifests after 2+ years of life[81]. Interestingly, accumulation of cytosolic DNA is associated with cellular aging and senescence with cGAS/STING implicated in these outcomes[82]. Whether the accumulation of cytosolic dsDNA and cGAS/STING activity is a cause or result of the aging process is an open question and warrants further study. Additionally, mutations in genes associated with the regulation of nuclear envelope structure and maintenance[83] as well as cGAS/STING[84] have been associated with the progression of neurodegenerative diseases such as Parkinson's Disease[85] and prion-mediated neurodegeneration[86]. Therefore, exploration of the role of the Zfp335/Ankle2/Baf pathway uncovered in this study may reveal important pathways regulating and potential therapeutic targets to mitigate, neurodegeneration, and cellular aging.

## Methods

### Ethics statement
All animal work performed in these studies was conducted in accordance with animal protocols approved by the Duke University Institutional Animal Care and Use Committee.

### Mice
B6.Cg-*Zfp335*[tm1Caw] (Zfp335[fl/fl], Stock No. 022413), B6.Cg-*Rag2*[tm1.1Cgn]/J (Rag2[−/−], Stock No. 8449), and B6J.129(B6N)-*Gt(ROSA)26Sor*[tm1(CAG-cas9*,-EGFP)Fezh]/J (R26[LSL-Cas9], Stock No. 026175) mice were purchased from The Jackson Laboratory. C57BL/6J-Tg(Cd8a*-cre)B8Asin (E8[III]-cre, Jax Stock No. 32080) mice[44] were generously provided by Jung-Hyun Park (NIH). B6.129S-Tcrd[tm1.1(cre/ERT2)Zhu] (*Tcrd*[CreERT2], Jax Stock No. 31679) have been maintained in our colony since original development[67]. A modified Ai6 targeting vector[87] to drive conditional overexpression of Bcl2 was generated by cloning in mouse *Bcl2* cDNA (Transomic Technologies) using FseI and SfiI restriction sites. R26[LSL-Bcl2] mice were generated by the Duke University Transgenic Facility using G4 mouse embryonic stem cells. Animals were maintained under specific pathogen-free conditions at the Cancer Center Isolation Facility of Duke University Medical Center. All experimental procedures were approved by the Duke University Institutional Animal Care and Use Committee. All mice used in this study were 4–8 weeks old. For all experiments, Cre-negative littermates were used as controls unless otherwise stated. Mice were euthanized by $CO_2$ inhalation followed by vital organ removal.

### Antibodies
All antibodies used in this study were purchased commercially and have previously been validated. Anti-TCRγδ (GL3) was purchased from BD Biosciences. Anti- TCRγδ (GL3), rabbit anti-Lamin B (10H34L18), polyclonal rabbit anti-Banf1 (Cat. PA5-20329), polyclonal rabbit anti-phospho IRF3 (Cat. PA5-36775), polyclonal rabbit anti-phospho STING (Cat. PA5-105674), donkey anti-rabbit IgG (H + L)-Alexa Fluor 488, goat anti-mouse IgG (H + L)-Alexa Fluor 488 and goat anti-rabbit IgG (H + L)-Alexa Fluor 647 were purchased from ThermoFisher Scientific. Anti-phospho TBK1 (D52C2) was purchased from Cell Signaling Technologies. Anti-CD16/32 (2.4G2) was purchased from Tonbo Biosciences. Anti-CD90.1 (OX7), anti-CD90.2 (30-H12), anti-CD4 (RM4-5), anti-CD8 (53-6.7), anti-CD44 (IM7), anti-CD25 (PC61), anti-CD62L (MEL-14), anti-TCRβ (H57-597), anti-CD27 (LG.3A10), anti-Bcl2 (BCL/10C4), anti-CD24 (M1/69), anti-B220 (RA3-6B2), anti-CD11b (M1/70), anti-CD11c (N418), anti-CD19 (6D5), anti-Ly6G/Ly6C (RB6-8C5), anti-NK1.1 (PK136), anti-TER119 (TER119), anti-CD117/c-kit (2B8), anti-Phosphoserine (M380B), mouse IgG1 isotype control (MG1–45), mouse IgG1 isotype control

(MOPC-21), anti-IFNAR-1 (MAR1-5A3), and Annexin V were purchased from Biolegend. Details for all antibodies used in this study can be found in Supplementary Table 1.

### Flow cytometry and cell sorting

Thymus or spleen tissues were harvested from 4–8-week-old mice. Tissues were then dissociated in FACS Buffer (PBS supplemented with 2.5% FBS and 2 mM EDTA) using a Dounce Homogenizer and filtered through 70 μm nylon mesh (Genesee Scientific) to yield single-cell suspensions. For spleen samples, red blood cells were lysed using 1× RBC lysis buffer and then resuspended in FACS buffer. $0.5–1 \times 10^7$ cells were stained with fluorescently labeled antibodies for 30 min at 4 °C, then washed with excess FACS buffer. Prior to analysis, propidium iodide (Sigma–Aldrich, Cat. P4170) or DAPI (Sigma–Aldrich, Cat. D9542) were added to a final concentration of 0.5 μg/mL or 100 ng/mL, respectively, for live/dead discrimination. Cells were analyzed on a Fortessa X20 (BD Biosciences) or FACSCantoII (BD Biosciences) cytometer. For isolation of thymocyte subsets or virally transduced cells, sorting was performed using a FACSDiva (BD Biosciences) or Astrios (Beckman-Coulter) cell sorter. For sorting of thymocyte subsets ex vivo, staining included a lineage dump stain consisting of B220, CD11b, CD11c, CD19, GR-1, NK1.1, TCRβ, TCRγδ, and TER119 antibodies. All analyses were performed using FlowJo v10 software (TreeStar). Detailed gating schemes are shown in Supplementary Fig. S11.

### ChIP-seq analysis

Previously published E2A[42] (GSE89849) was generated by our lab or Zfp335[46] (GSE58333) ChIP-seq datasets were accessed through the NCBI GEO database. Reads were aligned to the mm9 genome using Bowtie (version 1.1.2, parameters: -chunkmbs 128 -mm -m 1 -best -strata -p 4 -S -q). Peaks were called and Bed and wiggle files were generated for visualization using the Integrative Genome Browser using MACS2 (version 1.4.2, default parameters). Peaks were annotated using the NGS: Peak Annotation tool on Nebula.

### Bulk RNA-seq

DP thymocytes (Lin⁻ CD4⁺ CD8⁺) were FACS sorted from the total thymus of 7-week-old female Zfp335$^{fl/fl}$ E8$_{III}$-cre or Zfp335$^{+/+}$ E8$_{III}$-cre mice. Purified DP cells were lysed with Trizol and RNA isolated using the Direct-Zol Micro RNA prep kit (Zymo) according to the manufacturer's recommended protocol. gDNA was eliminated by on-column DNase digestion. Libraries were prepared using standard preparation protocols by BGI Genomics. 150 bp paired-end sequencing was performed on the BGISEQ-500 sequencing platform.

Paired-end reads were mapped to the mouse mm10 reference genome using the HiSat2 software, and count matrices were generated using the featureCounts function of the Subreads software package. Differential expression analysis was performed using DeSeq2 implemented through iDep.91 (http://bioinformatics.sdstate.edu/idep90/). For identification of differentially expressed genes $p$-values were adjusted based on a 10% false discovery rate, and genes with a fold-change of ≥2 and an adjusted $p$-value ≤0.05 were considered differentially expressed. Gene-Set Enrichment Analysis (GSEA) was utilized to identify enriched pathways based on differential expression analysis using pre-ranked gene lists with default parameters and the HALLMARKS gene sets (MSigDB). Pre-ranked gene lists were generated by multiplying the -log$_{10}$(p-value) by the direction of the fold-change.

### Cell culture

OP9-DL1 cells, kindly provided by Maria Ciofani (Duke University), were cultured in MEMα (Gibco) supplemented with 10% FBS (Atlanta Biologicals) and 1× penicillin/ streptomycin (Gibco). HEK293T cells were cultured in DMEM supplemented with 10% FBS, 1× penicillin/ streptomycin, 1× non-essential amino acids, and 1× GlutaMAX. For OP9-DL1 culture of thymocytes, cultures were additionally supplemented with 5 ng/mL recombinant mouse IL-7 (Biolegend). Scid.adh.2c2 cells were cultured in IMDM supplemented with 10% FBS (Hyclone), 1× penicillin/ streptomycin, 1× NEAA, 1× sodium pyruvate, 1× GlutaMAX, and 55 μM β-mercaptoethanol. In some OP9-DL1 cultures 0, 1, 10, 100, or 1000 nM rapamycin (Cayman Chemicals), 0, 1, 10, 100, or 1000 nM everolimus (Cayman Chemicals), 10 μg/mL anti-mouse IFNAR-1 antibody (Biolegend), 5 μg/mL RU.521 (Invivogen), 0.5 μg/mL H-151 (Cayman Chemicals) or 25 or 250 μg/mL Cridanimod (Cayman Chemicals) were added. All cultures were maintained at 37 °C with 5% $CO_2$.

### DN thymocyte enrichment

Total thymocytes were harvested from 4–8-week-old mice. Tissues were dissociated and strained through 30 μm nylon mesh (Genesee Scientific). For purification of DN3/4 thymocytes, cells were stained with biotinylated antibodies against B220, CD3, CD4, CD8, CD11b, CD11c, CD19, CD44, c-Kit, GR-1, IgM, NK1.1, TCRβ, and TCRγδ. For enrichment of total DN cells, CD44 and c-Kit antibodies were excluded. Following antibody staining, cells were incubated with 50 μL or 100 μL of streptavidin magnetic particles (Spherotech, cat. SVM-40-100) / 10⁷ cells at $2 \times 10^7$ cells/mL in FACS buffer for total DN enrichment or DN3/4 purification, respectively. Particle-bound cells were separated three times on a magnetic rack.

### Retrovirus packaging and transduction

Retrovirus was generated by transfecting HEK293T cells with 1 μg/mL each of MSCV transfer and pCL-Eco vectors using Lipofectamine 2000 (Invitrogen) or JetOptimus (Genesee Scientific) according to manufacturer's recommended protocols. Media was changed 24 h post-transfection and viral supernatants were harvested 24 h later. DN3/4-enriched thymocytes were transduced with fresh viral supernatant via spinfection for 2 h at $1065 \times g$ at 30 °C with 6.7 μg/mL polybrene (Millipore). Following spinfection cells were transferred to culture on OP9-DL1 stromal cells for overnight culture. 18–24 h post-infection virally transduced (DsRed + or Thy1.1+) DN3 (CD25+) or DN4 (CD25−) were isolated by FACS sorting for an additional 3–5 days of culture in the OP9-DL1 culture system. For dual-targeting CRISPR experiments, equal volumes of sgRNA-Thy1.1 and -DsRed viral supernatants were mixed for transduction.

### TCRα repertoire analysis

To prepare TCRα sequencing libraries, one million DP thymocytes were sorted from WT or Zfp335cKO thymuses, lysed with Trizol and RNA purified using the Direct-Zol Micro Prep kit (Zymo Research) according to the manufacturer's recommended protocol, including on-column DNase digestion. Sequencing libraries were prepared according to previously published methods[88,89]. Briefly, 5 ng of total RNA was reverse transcribed with Trac-RT primer and SMARTnnnA template switch oligo (IDT) using SmartScribe Reverse Transcriptase (Takara Bio). Following RT, samples were treated with 5U uracil DNA glycosylase (NEB) for 40 min at 37 °C. cDNA was purified using Sera-Mag Carboxylate-Modified Magnetic SpeedBeads (GE Healthcare Life Sciences). cDNA was amplified for 18 cycles using Q5 high-fidelity polymerase (NEB) and then purified with Sera-Mag SpeedBeads. Amplified cDNA was then dual-indexed by PCR with Q5 polymerase and purified again. Following indexing samples were quality-controlled by agarose gel electrophoresis, pooled at equal concentration, and final libraries prepared using the NEBNext Ultra II DNA library preparation kit (NEB). Prior to sequencing, libraries were gel extracted using the Zymoclean Gel DNA recovery kit (Zymo Research). All primer sequences can be found in Supplementary Table 2. 300 × 300 bp sequencing was performed on a MiSeq sequencer (Illumina).

TCRα sequencing analysis was performed as follows. First, raw fastq files were demultiplexed using the Checkout function of Migec

v1.2.9[90] with the following parameters: -cute. Molecular identifier group (MIG) size distributions were determined using the Histogram function of Migec with default parameters. Based on these output UMIs were collapsed and filtered using the AssembleBatch function of Migec with the following parameters:–force-collision-filter –force-overseq 4. Collapsed reads were then merged using MiTools v1.5 (https://github.com/milaboratory/mitools) with the following parameters: -ss -s 0.7. Next, reads were aligned to the mouse TCRα locus using the align function of MiXCR v2.1.1[91] with the following parameters:–loci TRA -s mmu -OvParameters.geneFeatureToAligh = VTranscript. Aligned reads were then used to assemble clonotypes using the assemble function of MiXCR with the following parameters: -OassemblingFeatures = VDJRegion, then exported for downstream analysis using exportClones function of MiXCR. Gene segment usage was determined using the CalcSegmentUsage function of VDJTools v1.2.1[92] with the following parameters: -u. Gene segment usage was manually sorted based on genomic coordinates.

## scRNA-seq library preparation
For single-cell RNA-sequencing, DN4 thymocytes (Live Lin⁻ CD4⁻ CD8⁻ CD25⁻ CD44⁻) were sorted from one male and one female mouse pooled for each genotype using an Astrios Sorter. Sorted cells were encapsulated into droplets and libraries were prepared using a Chromium Single Cell 3′ Kit using the v3.1 chemistry. Seven thousand cells per genotype were targeted. scRNA-seq libraries were pooled and sequenced on a NovaSeq S Prime Flow Cell yielding an average depth of 71,584 or 67,816 reads per cells for Zfp335cKO or WT samples, respectively.

## scRNA-seq analysis
scRNA-seq data were processed using the Cell Ranger pipeline (10x Genomics). FASTQ files were generated from raw base call logs (bcl2fastq, v2.20), aligned to the mouse mm10 (release 93) reference genome (cellranger, v3.1.0; STAR v2.5.3a) to generate raw gene count matrices. To ensure coverage of all relevant genes in our dataset, protein-coding, lincRNA, antisense, and all immunoglobulin and T-cell receptor V, D, J, and C genes, including pseudogenes, were used for the generation of the mm10 annotation file.

All downstream analyses were performed using the R software package Seurat (v4.0.0). Data were filtered to exclude cells with <1000 genes detected or <1000 UMIs. Doublets were excluded by filtering cells with >60,000 UMIs. Low-quality cells were further filtered by removal of cells with >7.5% mitochondrial gene expression. Gene expression matrices were then merged using the integration anchor method in Seurat based on the 6000 most variably expressed genes, log normalized with a scaling factor of 10,000, scaled using variance stabilizing transformation and cell cycle phase determined using the CellCycleScoring function in Seurat with the built-in S phase and G2M phase gene lists. Dropouts were imputed using the R package ALRA. Imputed data were only used for the determination of the frequency of gene expression and not differential expression analysis. The cell cycle phase was regressed, and principal component analysis (PCA) was performed on the 6000 most variable genes. 35 principal components were selected for downstream analysis based on JackStraw analysis. Nearest neighbor graphs were constructed based on the first 35 principal components using the FindNeighbors functions in Seurat. Clusters were determined using FindClusters with a resolution of 0.5. Dimensionality reduction was performed by Uniform Manifold Approximation and Projection (UMAP) using 35 principal components. Gene expression was visualized by VlnPlot, DotPlot, and FeaturePlot functions in Seurat. Gene signature scores were calculated using SingleCellSignatureExplorer and previously described methods[93]. Differential expression analysis was performed using the FindAllMarkers or FindMarkers functions in Seurat on the normalized count matrix with Wilcoxon Rank Sum Test (parameters: assay = 'RNA', logfc.threshold = 0).

## Cloning cDNA overexpression vectors
Bcl2 overexpression vector was generated by cloning Bcl2 cDNA (Transomic Technologies, Cat. TCM1304) into the pMSCV-loxp-dsRed-loxP-eGFP-puro-WPRE vector (Addgene #32702) using the EcoRI and NsiI restriction sites. Ankle2 cDNA (Transomic Technologies, Cat. TCM1004) was cloned into the MSCV-IRES-Thy1.1 vector using NEBuilder Hifi Assembly (New England Biolabs). All vectors were propagated in Stbl3 cells (ThermoFisher Scientific).

## Generation of *Scid.adh.2c2-dCas9*[10x−GCN4] CRISPRi cells
dCas9[10x-GCN4] (pHRdSV40-dCas9-10xGCN4_v4-P2A-BFP, Addgene #60904) was lentivirally transduced into Scid.adh.2c2 cells, following which BFP + cells were isolated by flow cytometry. Single cells were then cloned into 96-well plates and screened for knockdown efficiency using CD25 gRNA retroviral vectors. Clones exhibiting more than 90% CD25 downmodulation were expanded for use in our studies.

## Generation of gRNA retroviral vectors
All gRNAs were designed using the CRISPick[94] gRNA design tool. All gRNAs were cloned into expression vectors by annealing followed by ligation into a BbsI cleavage site. The basic gRNA expression vector used was the MSCV-mU6-sgRNA-hPGK-Thy1.1 (kindly provided by Maria Ciofani). Knockout gRNAs were first cloned into this Thy1.1 backbone. To generate DsRed expressing vectors for dual targeting, Thy1.1 was removed by digestion with BamHI and EcoRI and replaced with DsRed Express II by NEBuilder Hifi Assembly. The CRISRPi retroviral vector was generated by first cloning the pSV40-scFv-GCN4-sfGFP-VP64-GB1-NLS (Addgene #60904) fusion construct into the MSCV-mU6-sgRNA-hPGK backbone followed by replacement of VP64 with KRAB using NEBuilder.

## qPCR analysis of gene expression
Following viral transduction, Scid.adh.2c2.dCas9[10x-GCN4] cells were assessed for transduction efficiency by flow cytometry. For samples exceeding 90% GFP +10⁶ cells were lysed in Trizol and RNA isolated using the Direct-Zol MicroPrep kit. 500 ng of RNA was reverse transcribed using SuperScript III Reverse Transcriptase (Invitrogen) with random hexamers according to the manufacturer's recommended protocol. 5 ng of cDNA per 25 μL reaction was then used for gene expression analysis with PowerTrack Sybr Green Master Mix (Applied Biosciences) according to the manufacturer's recommended protocol using fast cycling conditions with an Eppendorf MasterCycler qPCR machine. Relative expression was determined using the ddCt method, with Gapdh being used for normalization.

## Proximity ligation assay for Baf phosphorylation
Proximity ligation assays were performed using the Duolink® flow-PLA Detection Kit – Orange (Millipore Sigma, Cat. DUO94003) according to manufacturer's recommended protocol with minor changes. Briefly, total thymocytes were prepared as described in the flow cytometry and cell sorting methods section. 10⁷ thymocytes were stained with surface antibodies to distinguish all major thymocyte subsets. Next, cells were fixed with 4% paraformaldehyde for 10 min, washed, and permeabilized for 30 min at room temperature. After permeabilization, cells were blocked for 1 h at 37 °C with 300 μl of Duolink® blocking solution, then stained overnight at 4 °C with purified mouse anti-Phosphoserine (Biolegend) or purified mouse IgG1 isotype control (clone MG1–45, Biolegend) and purified rabbit anti-Banf1 (ThermoFisher Scientific) or rabbit anti-Lamin B1 (ThermoFisher Scientific) diluted in Duolink® antibody diluent. After each step, cells were washed twice with 1 mL or Duolink® In Situ wash buffer. Next, cells were incubated for 1 h at 37 °C with Duolink® In Situ PLA® Probe Anti-Rabbit PLUS (Sigma Millipore, Cat. DUO92002) and Duolink® In Situ PLA® Probe Anti-Mouse MINUS (Sigma Millipore,

Cat. DUO92004) diluted in Duolink® antibody diluent. Additional controls in which individual probes were omitted were also prepared. Following probe incubation, cells were washed and then incubated for 30 min at 37 °C with 1× Duolink® ligation reaction mixture, washed again, and incubated for 90 min with 1× Duolink® amplification reaction mixture. Following amplification, cells were incubated with 1× Duolink® Detection Solution–Orange for 15 min at 37 °C. Cells were finally washed, resuspended in PBS, and assayed using a FortessaX20 cytometer (BD Biosciences).

### Determination of nuclear envelope structure and measurement of cytosolic dsDNA

$5 \times 10^4$ Hela cells per well were reverse transfected with 15pmol siRNA using Lipofectamine RNAiMax (ThermoFisher Scientific) in an eight-well chamber slide according to recommended protocols. ANKLE2 and universal nontargeting control siRNAs were purchased from IDT (Design ID: hs.Ri.ANKLE2.13). BANF-targeting siRNAs were purchased from ThermoFisher Scientific (IDs: s16807, s16808, 26065). Forty-eight hours post-transfection cells were fixed with 4% paraformaldehyde for 10 min at room temperature and permeabilized with permeabilization buffer for 1 h at RT temperature. Primary antibody Lamin B (Invitrogen, Cat. 702972) were added for overnight incubation at 4 °C and washed with 1× PBS for three times. After that, the secondary antibody Alexa Fluor 647-conjugated goat anti-rabbit antibody (Invitrogen, Cat. A32733) were added for 12 h at 4 C in the dark. After washing with 1× PBS for three times, slides were mounted with DAPI-containing mounting media (VECTORLAB, Cat. H-1200). Images were collected using Zeiss 780 upright confocal. To analyze nuclear structure DAPI channel images were converted to binary with ImageJ. Following binarization, the Watershed function was used to separate touching cells. Circularity was then determined with a minimum threshold of 500 px$^2$.

For analysis of ex vivo DN4 thymocyte nuclear envelope DN4 cells were isolated by magnetic bead-based purification. Following purification, cells were fixed with 4% paraformaldehyde (PFA) for 10 min at room temperature. Cells were then stained overnight with purified rabbit anti-Lamin B1 antibody, followed by incubation with goat anti-rabbit IgG AlexaFluor 488 secondary antibody (Thermo-Fisher Scientific) for 1 h. Cells were spun onto slides using the CytoSpin4 centrifuge (ThermoFisher Scientific) and mounted with ProLong™ Gold Antifade Mountant with DAPI (ThermoFisher Scientific). For assessment of cytosolic dsDNA, DN3/4 thymocytes were isolated and expanded in OP9-DL1 culture for 3 days. Cells were harvested from culture, fixed with 4% PFA for 10 min at room temperature, and permeabilized for 5 min with 0.01% Triton X-100. Cells were stained overnight with purified mouse anti-dsDNA antibody (Clone AE-2, ThermoFisher Scientific, followed by incubation with donkey anti-mouse IgG AlexaFluor 488 secondary antibody and rat anti-mouse CD90.1 AlexaFluor 647 for 1 h. Cells were adhered to slides and mounted with ProLong™ Fold Antifade Mountant with DAPI. For assessment of IRF3 translocation, cells were prepared identically to the experiments measuring cytosolic dsDNA but stained with rabbit anti-mouse phosphor-IRF3 (ThermoFisher Scientific) primary and goat anti-rabbit IgG AlexFluor 488 secondary and rat anti-mouse CD90.1 AlexFluor 647 antibodies.

Microscopy images were acquired on the Zeiss 710 Inverted Laser Scanning Confocal Microscope (Duke University Light Microscopy Core Facility). For quantification, at least 20 cells per animal were imaged as z-stacks using the 63x oil immersion objective. Image quantification was conducted with the Imaris for Neuroscientists Cell Imaging Software v. 9.3.0 (Bitplane) using the Surfaces tool on the acquired DAPI signal to identify nuclei. Identified nuclei were then differentiated by the mean fluorescent intensity of Lamin B1, and quantified. The percentage of nuclei identified as Lamin B1$^{high}$ and the mean sphericity of all nuclei for each animal were used as a single n for statistical analysis. Additionally, mean DAPI intensity was quantified for individual nuclei, with each nucleus represented as a single data point. For analysis of cytosolic dsDNA, the Surfaces tool was used to define nuclei based on acquired DAPI signal, and cell membrane was defined based on acquired CD90 signal. dsDNA was defined using the vesicles tool and cytoplasmic staining defined automatically. Within each experiment, dsDNA abundance was determined for each individual cell and normalized to the average of all control cells in the experiment.

### 2′,3′-cGAMP ELISA

To measure total cellular cGAMP DN3/4 thymocytes were expanded in OP9-DL1 culture for 3 days. Cells were then harvested, washed with PBS, and lysed in 100 μl MPER buffer (ThermoFisher Scientific) for 15 min at room temperature. Lysates were cleared by centrifugation at $16,000 \times g$ for 15 min. Total protein content for lysates was determined using Pierce Detergent-Compatible Bradford Assay (ThermoFisher Scientific) for normalization of cGAMP. cGAMP ELISAs performed using the 2′,3′-cGAMP ELISA (Cayman Chemicals) according to the manufacturer's recommended protocol.

### In vivo H-151 treatment of mice

Mice were administered 750 pmol (210 μg) of H-151 (Cayman Chemicals) or vehicle via intraperitoneal injection daily for 7 days beginning at 7 weeks of age. The vehicle for injections was sterile PBS + 10% Tween-80 (VWR).

### In vivo generation of post-β-selection Rag2$^{−/−}$ thymocytes

Nine-week-old male and female Rag2$^{−/−}$ mice were administered 150 μg of ULTRA-LEAF purified anti-CD3 (Biolegend) via intraperitoneal injection. Ten days post-injection thymi were harvested and subjected to overnight stimulation with STING agonist followed by assessment of apoptosis by Annexin V binding.

### Statistical analysis

Statistical tests were performed using GraphPad v9.0.0 (Prism). For graphs with multiple comparisons being made, two-way ANOVA was performed with post hoc Sidak's test or Tukey's test for multiple comparisons. For comparisons of cell numbers, data were log transformed prior to statistical tests. For all Two-way ANOVA tests, normality tests were performed to ensure normalcy assumptions were met. For graphs of single comparisons, a two-tailed Mann–Whitney test was used. All significant $p$-values are shown in each graph. For all experiments, data are reported based on individual animals or biological replicates pooled from multiple animals. No reported data are from repeated measurements of the same samples. No statistical methods were used to predetermine the sample size.

### Reporting summary

Further information on research design is available in the Nature Research Reporting Summary linked to this article.

## Data availability

Data generated in this study can be accessed upon publication through NCBI Gene Expression Omnibus (https://www.ncbi.nlm.nih.gov/geo/) under accession GSE189244 for bulk and scRNA-seq or GSE205323 for TCRα repertoire analysis. All data are included in the Supplementary Information or available from the authors upon reasonable requests, as are unique reagents used in this Article. The raw numbers for charts and graphs are available in the Source Data file whenever possible. Source data are provided with this paper.

## Code availability

All codes from this study are available through GitHub (https://github.com/ratiujer/Zfp335-scRNAseq).

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

## Acknowledgements

This study was funded by the NIH (R01-GM059638 and P01-AI102853) to Y.Z., (P01-AI102853) to D.W., and (RO1-AI099100, RO1-NS120417, and RO1-AI160737) to M.L.S. We thank M. Cook, N. Martin, B. Li, and L. Martinek (Duke University Cancer Institute Flow Cytometry Core) for technical support and cell sorting. We thank the Duke Molecular Physiology Institute for preparation of scRNA-seq libraries. We thank M. Krangel, QJ Li, J. Racine, D. Serreze, and M. Hasham for critical reading and comments on the manuscript. We thank M. Ciofani and J. Park for providing cell lines and mice. We thank M. Parker and J. Wheaton of M. Ciofani's lab for providing MSCV-Thy1.1 and MSCV-sgRNA expression vectors.

## Author contributions

J.J.R., M.L.S., D.W., and Y.Z. designed experiments, and analyzed and interpreted data. J.J.R., W.E.B., E.L., Q.W., N.M., D.D., M.J.H., S.W., S.R., and A.V.C. performed experiments and analyzed data. J.J.R., Y.Z., and D.W. wrote the manuscript with editing by the co-authors. J.J.R. and Y.Z. oversaw and supervised all aspects of the study.

## Competing interests

The authors declare no competing interests.
