## [Peer Review File · Nature Communications]

REVIEWER COMMENTS

Reviewer #1 (Remarks to the Author):

The authors conducted a functional analysis of the transcription factor Zfp335 in DN4 thymocytes and uncovered an unexpected link between the Zfp335 and cGAS-STING axis in DN4 thymocytes. The authors generated Zfp335^{fl/fl} E8III-cre mouse and found that the genetic deletion of Zfp335 led to a significant reduction in DN4 thymocytes. Using several cell biological experiments, authors showed that Zfp335 cKO thymocytes are dying during the post-beta-selection proliferative phase, and TCRbeta rearrangement and subsequent pre-TCR expression are unimpaired in Zfp335 cKO cells. Authors also performed scRNA-seq to assess whether there is any heterogeneity in the DN4 compartment exacerbated by Zfp335 deficiency and defined the 'true' DN4 thymocyte population. Furthermore, authors found that the Ankle2-Baf axis is involved in the cGAS-STING axis dependent apoptosis of DN4 thymocytes. This study also showed the mechanism of STING-mediated induction of apoptosis by loss of Zfp335 in an in vivo model. It is also novel to explore the STING pathway associated with the upstream signaling of Bcl2. While this study focuses an interesting topic and is a well-designed, it seems necessary to emphasize the novelty of the regulation of the cGAS-STING pathway by Zfp335.

Main Comments in no specific order:

1. Is Zfp335 expressed specifically in DN4 thymocytes? Also, how is the expression kinetics of Zfp335 during thymic T cell development? If Zfp335 is ubiquitously expressed in thymocytes, why Zfp335 is selectively critical for DN4 stage but not DN2 stage? A rational explanation is needed.
2. In the exploration of true DN4 thymocytes, are there any cell populations that express high levels of ISGs or type-I IFN-related genes? Related to this, the frequency of Mat2 cell population was particularly affected by the loss of Zfp335 (Figure 4C). Would be better with a discussion on this issue.
3. Since the regulation of the cGAS-STING pathway by the loss of Zfp335 will be the novel point of this paper, further molecular analysis is required. In particular, as a marker for activation of the cGAS-STING pathway, author should perform whether the phosphorylation or nuclear translocation of STING is altered by the deletion of Zfp335 (either is sufficient). The phosphorylation and nuclear translocation of IRF3 also need to be investigated.
4. Authors also mentioned that the loss of Ankle2 changes the structure of the nuclear membrane, which is important for the activation of cGAS-STING. Does the amount of cytosolic DNA and cGAMP increase? These can be addressed by ELISA, DNA quantification, and anti-DNA antibody labeling by microscopy. If any accumulation was not found, does deficient of Zfp335 change the sensitivity of STING to ligands?
5. The cGAS-STING pathway is known to control mTOR activity in addition to IRF3-dependent induction of I-IFN. Which downstream signals of STING are critical in the induction of apoptosis by the deletion of Zfp335? mTOR inhibitor and neutralizing antibodies to IFNAR can be used to investigate.
6. The authors discussed the possibility that Zfp335/cGAS/STING-dependent induction of apoptosis may be involved in diseases such as zika virus infections and CNS immune cell-related diseases. Is the mechanism of apoptosis induction found in this study also involved in more diverse biological phenomena such as aging?

Minor Concerns with data and/or data presentation:

- a. It is necessary to describe in detail how the classification by cell cycle in Figure 4B was performed.
- b. The legend in Figure 2 is needed to include the explanation of H.
- c. The legend in Figure 3 is needed to include the explanation of C and D.

- d. Figure 4E is needed to be annotated in the main text.
- e. The legend descriptions for Figure 5A and 5E are needed to be confirmed.
- f. Annotations of Fig S4E and Fig S4F-I in line 267 and 269 are needed to be confirmed. Figure S5 may be correct.
- g. The sentence that begins "Ankle2 has previously..." in line 305 is required to cite a reference.

Reviewer #2 (Remarks to the Author):

By using multiomics sequencing, the authors demonstrated that Zfp335 deficiency led to defective T cell development via dysregulation of the Zfp335/Ankle2/Baf axis, which ultimately drives cGAS/STING-mediated DN4 cell death. They have also suggested that DN4 cells are heterogeneous and defined a "true"DN4 cell population. However, while these findings are important and the conclusions are supported by solid data, a paper published very recently obtained similar conclusion that Zfp335 is an important regulator of early T cell development in the thymus via modulating apoptosis (Wang, Jiao, Sun, et al. eLife 2022;11:e75508). Thus, the novelty of the current study is a concern.

Questions:

1. What is the expression profile of Zfp335 and Ankle2 during different T cell development stages? whether Zfp335 deficiency also regulates apoptosis of DP thymocytes?
2. Inactivation of pre-TCR signals dampens T-lymphocyte development by arresting thymocytes at the DN3 stage and inducing apoptosis (Fehling et al., 1995). Does Zfp335 deficiency regulate apoptosis via regulating pre-TCR activation?
3. What determines the unique sensitivity of DN4 cells to STING-mediated apoptosis?
4. There are 5 clusters of "True"DN4 cells. Whether Zfp335 plays different roles in apoptosis among these clusters?

Reviewer #3 (Remarks to the Author):

The paper reports the role of transcription factor Zfp355 in the DN4 thymocyte development stage. I was tasked to review systems biology and computational biology aspects of the paper. While overall the computational analyses of the data look reasonable, I have major concerns that need to be resolved before publication (or in fact before I can reasonably assess the computational analyses).

Major comments:

- No data reported in the paper (bulk RNA-seq and scRNA-seq for WT and Zfp355-KO cells) was provided. All the high-throughput data reported in the manuscript should be available for review in order to establish reproducibility and enable some validation. The paper states all data will be available at a certain GEO accession number, but currently is it not.
- Exposition of the data analysis in the main figures, figure captions, and in Methods are sometimes vague, and more detailed description of the methodology and content of the figures is warranted.

- I have also potential concerns / questions about some pieces of computational analysis where additional alternative analyses and/or parameters are needed to better establish robustness of observations.

Specifically:

- Fig 1J,K, Line 622: edgeR and DESeq2 were used for differential expression analysis. But how exactly? With which parameters? Typically the whole task of differential expression analysis is performed either entirely using edgeR, or entirely using DESeq2. It is not clear how the two methods were combined here.

- Fig 1L: Same with GSEA analysis (Line 623). Not clear what parameters were used, and what exact lists of genes, coming from the previous step, were used. Not clear what exactly is shown in Fig 1L -- all enriched terms? Manually selected ones? Were all differentially expressed genes used for GSEA analysis or only the downregulated ones?

- Fig 1J: caption says "differential expression of select Zfp355-target genes". Does this mean this differential expression analysis was restricted only to Zfp355 targets? If yes, then how were these targets defined? What is the motivation for this?

- Fig 1K: Some ChIP-seq data was used. But not clear how exactly it was analyzed. This is not explained in Methods.

- The authors used cellranger, apparently with default parameters and standard mm10 mouse reference genome, to preprocess the data and obtain raw gene expression matrices. How accurate this is with respect to T cell receptor genes in their particular cells vs. these loci in the reference? How accurate are Trac, Trbc1, Tcfg-C1, etc. gene expression estimates obtained in this manner?

- Line 677: data was normalized, scaled, cell cycle scored -- how exactly and with which parameters?

- Line 678: dropouts were imputed. Does this mean that the subsequent analysis uses imputed rather than raw expression values? It is controversial in the field if it is appropriate to rely on imputed scRNA-seq gene expression values for anything other than just visualizations of expression. Therefore I suggest that the authors repeat some of their major analytical steps without imputation, thus confirming their main results.

- Line 682: UMAP and clustering -- the text implies that clustering was performed after UMAP, on UMAP-embedded (2-dimensional?) data. If this is the case, it is incorrect. Clustering should be performed on higher dimensional data, i.e. in this case, on 35-dimensional PCA. This should be clarified in the text of Methods. Also, line 683 refers to clustering with parameter 0.5, but what algorithm or Seurat function does this parameter refer to?

- Fig 4C: How exactly were these frequencies calculated? i.e. frequencies of what in what are these values?

- Fig S5B,C: How exactly differential expression analysis was run in these plots?

- Fig 5: caption doesn't correspond to figure. Probably plot correspond to caption (A) is not in the figure.

RESPONSE TO REVIEWER COMMENTS

Reviewer #1 (Remarks to the Author):

The authors conducted a functional analysis of the transcription factor Zfp335 in DN4 thymocytes and uncovered an unexpected link between the Zfp335 and cGAS-STING axis in DN4 thymocytes. The authors generated Zfp335^{fl/fl} E8III-cre mouse and found that the genetic deletion of Zfp335 led to a significant reduction in DN4 thymocytes. Using several cell biological experiments, authors showed that Zfp335 cKO thymocytes are dying during the post-beta-selection proliferative phase, and TCRbeta rearrangement and subsequent pre-TCR expression are unimpaired in Zfp335 cKO cells. Authors also performed scRNA-seq to assess whether there is any heterogeneity in the DN4 compartment exacerbated by Zfp335 deficiency and defined the 'true' DN4 thymocyte population. Furthermore, authors found that the Ankle2-Baf axis is involved in the cGAS-STING axis dependent apoptosis of DN4 thymocytes. This study also showed the mechanism of STING-mediated induction of apoptosis by loss of Zfp335 in an in vivo model. It is also novel to explore the STING pathway associated with the upstream signaling of Bcl2. While this study focuses an interesting topic and is a well-designed, it seems necessary to emphasize the novelty of the regulation of the cGAS-STING pathway by Zfp335.

Main Comments in no specific order:

1. Is Zfp335 expressed specifically in DN4 thymocytes? Also, how is the expression kinetics of Zfp335 during thymic T cell development? If Zfp335 is ubiquitously expressed in thymocytes, why Zfp335 is selectively critical for DN4 stage but not DN2 stage? A rational explanation is needed.

The authors thank the reviewer for all of their insightful comments and questions. Zfp335 is ubiquitously expressed throughout T cell development with peak expression occurring in DP cells (Fig S1C). We did not intend to suggest that Zfp335 is only critical for the DN4 stage of T cell development. It is reasonable to hypothesize that it may also play key role(s) in other stages of development including DN2. However, in our studies the Cre used to mediate Zfp335 deletion is not active until the DN3a stage. Therefore, we are unable to assess the role for Zfp335 in DN2 cells without breeding a new strain. We have added clarified this point in the discussion.

"Our studies highlight the necessity of sustained Zfp335 and Ankle2 expression to promote DN4 survival and support proper T cell development. Zfp335 is ubiquitously expressed throughout T cell development. Therefore, it is likely that Zfp335 plays numerous roles throughout thymopoiesis. In addition to supporting DN4 survival we show that it is also

required for terminal thymocyte maturation. Based on the mechanism uncovered in our work and the high degree of functional similarity between DN4 and DN2 cells it is likely that loss of Zfp335 at the DN1 stage or earlier may block T cell development at the DN2 stage. Such a hypothesis is support by a recent study which found that mice carrying a human disease associated STING mutation (V154M) exhibit severe T lymphopenia likely due to increased rates of early thymic progenitor apoptosis⁶³. However, to test this hypothesis alternative means of Zfp335 deletion will need to be utilized.”

2. In the exploration of true DN4 thymocytes, are there any cell populations that express high levels of ISGs or type-I IFN-related genes? Related to this, the frequency of Mat2 cell population was particularly affected by the loss of Zfp335 (Figure 4C). Would be better with a discussion on this issue.

In analyzing our scRNA-seq data we did not find any particular population of ‘true’ DN4 cells that express particularly high levels of ISGs or IFN-I-related genes at the gene signature level. This data has been added as Supplementary Figure 9A in relation to Reviewer #1 comment 5. We did however find elevated IFN-I gene signatures among Zfp335cKO Mat T cells (Fig S6). Based on these data along with data related to Reviewer #1 comment 5 it seems unlikely that IFN-I signaling is playing a significant role in DN4 cells from Zfp335cKO mice. However, the elevated IFN-I gene signature among DP cells (from our bulk RNA-seq

dataset) and Mat populations (scRNA-seq) suggests that this signaling pathway may influence later stages of T cell development. The following discussion of this topic has been added to discussion section:

“ Consistent with studies of cGAS/STING signaling in mature T cells^{30, 33}, we found cGAS/STING-mediated apoptosis in DN4 cells is independent of IFN-I signaling. Interestingly, we found that Zfp335-deficient DP and DN4-like maturing T cells exhibit increased IFN-I signaling activity compared to controls. Similar transcriptional activity was previously observed in *Zfp335^{bloto}* mice⁴². While tonic IFN-I signaling is critical to normal T cell development⁶⁸, enhanced signaling has been shown to severely impair thymopoiesis⁶⁹. The mechanism by which Zfp335 and Ankle2 regulate terminal T cell maturation in the thymus is unclear. However, it is possible that enhanced IFN-I signaling resulting from cGAS/STING activation may contribute to this defect.”

Supplementary Figure 9A. Type I Interferon signaling gene signature among the 8 clusters of DN4 cells identified from analysis of WT (grey) or ‘true’ mutant Zfp335cKO DN4 cells (red).

A REACTOME-INTERFERON-ALPHA-BETA-SIGNALING**B** REACTOME-INTERFERON-GAMMA-SIGNALING
Supplementary Figure 6. Type I Interferon signaling (A) and interferon γ signaling (B) gene signature among all phenotypically defined DN4 cells identified from analysis of WT (grey) or mutant Zfp335cKO (red). P-values shown are the adjusted p-values for comparison of WT and Zfp335cKO cells within each cluster as determined by Wilcoxon Rank Sum tests.

Regarding what distinguishes the Mat_2 population from the other Mat populations and how this might explain the selective reduction of this population in the Zfp335cKO sample we compared gene expression between each population. We found that Mat_2 specifically lacks expression of Cd24a (encoding CD24, Fig 4E), loss of which coincides with terminal T cell maturation. These data have been added to Fig 5E. We showed that loss of Zfp335 leads to a secondary deficiency in the ability of these cells to undergo terminal maturation (Fig S5A-C). Based on this finding we believe the Mat_2 population are cells which have progressed from immature to mature. Therefore, we conclude that the selective reduction in the Mat_2 population observed in our scRNA-seq dataset is the result of the secondary terminal differentiation defect resulting from loss of Zfp335. Discussion of this point was added to the results section as follows:

“Additionally, we found that Mat_2 cells lack expression of CD24 (Fig 4E) suggesting that this population may represent mature thymocytes explaining the selective reduction in proportions of these cells in Zfp335cKO samples.”

Figure 4E – Expression of positive selection-associated genes and Cd24a across our full scRNA-seq dataset.

3. Since the regulation of the cGAS-STING pathway by the loss of Zfp335 will be the novel point of this paper, further molecular analysis is required. In particular, as a marker for activation of the cGAS-STING pathway, author should perform whether the phosphorylation or nuclear translocation of STING is altered by the deletion of Zfp335 (either is sufficient). The phosphorylation and nuclear translocation of IRF3 also need to be investigated.

We assayed the phosphorylation status of STING, TBK1, and IRF3 in WT and Zfp335cKO thymocytes. We found significantly increased phosphorylation of each protein in Zfp335cKO cells compared to control (Fig 6O-P). Additionally, we validated pIRF3 nuclear translocation by confocal microscopy (Fig 6Q).

4. Authors also mentioned that the loss of Ankle2 changes the structure of the nuclear membrane, which is important for the activation of cGAS-STING. Does the amount of cytosolic DNA and cGAMP increase? These can be addressed by ELISA, DNA quantification, and anti-DNA antibody labeling by microscopy. If any accumulation was not found, does deficient of Zfp335 change the sensitivity of STING to ligands?

We found that loss of Zfp335 leads to a significant accumulation of cytosolic dsDNA (Fig 6H-I), as well as, increased concentrations of cGAMP (Fig 6N) in thymocytes derived from 3 day OP9-DL1 culture seeded with DN3/4 cells.

Figure 6 – Representative maximum intensity projections (H) and quantification (I) of cytosolic dsDNA in WT or Zfp335cKO thymocytes. For quantification cytosolic dsDNA abundance was calculated relative to the mean abundance for WT cells. (N) Quantification of total cGAMP levels in WT or Zfp335cKO thymocytes normalized to protein content of the lysate used.

5. The cGAS-STING pathway is known to control mTOR activity in addition to IRF3-dependent induction of I-IFN. Which downstream signals of STING are critical in the induction of apoptosis by the deletion of Zfp335? mTOR inhibitor and neutralizing antibodies to IFNAR can be used to investigate.

IFNAR-1 blocking antibody or the mTOR inhibitors rapamycin and everolimus had no effect on rates of Zfp335cKO cell death. These data have been added to Supplementart Figure 9C and D-E, respectively. Related to Reviewer #1 comment 2 we examined our sequencing datasets for transcriptional activity downstream of IFN-I and mTOR signaling. We did not observe substantial changes in activity for either pathway between Zfp335cKO and WT DN4 cells (Fig S9A-B).

Supplementary Figure 9 – IFN-I (A) and mTOR (B) signaling gene signatures among the 8 clusters identified in our analysis of ‘true’ Zfp335 mutant DN4 cells. Quantification of frequency of apoptosis among thymocytes generated by 3 day OP9-DL1 culture in the presence of IFNAR-1 blocking antibody (C), or mTOR inhibiting small molecules rapamycin or everolimus (D,E).

6. The authors discussed the possibility that Zfp335/cGAS/STING-dependent induction of apoptosis may be involved in diseases such as zika virus infections and CNS immune cell-related diseases. Is the mechanism of apoptosis induction found in this study also involved in more diverse biological phenomena such as aging?

It is possible that similar mechanisms to what we have uncovered here may contribute to many biological phenomena such as aging. Accumulation of cytosolic DNA is associated with aging and neuropathologies such as Parkinson’s Disease however, many reports point to mtDNA as the source of cGAS ligand. Interestingly, mutations in BANF1 are known to cause Nestor-Guillermo Progeria Syndrome which is characterized by severe premature aging. Based on this, we have expanded our discussion of the pathway involving cell death downstream of Zfp335/Ankle2/Baf via cGAS/STING to include these additional topics as follows:

“Beyond CNS development the mechanism uncovered through or studies may apply to more diverse biological phenomena. Mutations in BAF have previously been shown to drive Nestor-Guillermo Progeria Syndrome, a disease associated with severe pre-mature aging which typically manifest after 2+ years of life⁷⁶. Interestingly, accumulation of cytosolic DNA is associated with cellular aging and senescence with cGAS/STING implicated in these outcomes⁷⁷. Whether the accumulation of cytosolic dsDNA and cGAS/STING activity is a cause or result of the aging process is an open question and further study warranted. Additionally, mutations in genes associated with the regulation of nuclear envelope structure and maintenance⁷⁸ as well as cGAS/STING⁷⁹ have been associated with progression of neurodegenerative diseases such as Parkinson’s Disease⁸⁰ and prion-mediated neurodegeneration⁸¹. Therefore, exploration of the role for the Zfp335/Ankle2/Baf pathway uncovered in this study may reveal novel pathways regulating, and potential therapeutic targets to mitigate, neurodegeneration and cellular aging.”

Minor Concerns with data and/or data presentation:

a. It is necessary to describe in detail how the classification by cell cycle in Figure 4B was performed.

Cell cycle classification was performed in the R package Seurat using the CellCycleScoring function. This function uses a built-in set of genes to define S phase and G2M phase of the cell cycle. The program uses these gene sets to assign a score for S phase and G2M phase genes which are then used to define cell cycle phase. This method is the standard means of defining cell cycle phase using Seurat.

b. The legend in Figure 2 is needed to include the explanation of H.

The figure legend has been updated to include a description of H.

c. The legend in Figure 3 is needed to include the explanation of C and D.

The figure legend has been updated to include a description of C and D.

d. Figure 4E is needed to be annotated in the main text.

Figure 4E has been annotated in the main text.

e. The legend descriptions for Figure 5A and 5E are needed to be confirmed.

The figure legend has been updated to properly annotate panels A and E.

f. Annotations of Fig S4E and Fig S4F-I in line 267 and 269 are needed to be confirmed. Figure S5 may be correct.

Annotations for Fig S4E-H have been updated and confirmed. These panels are now part of Fig S6.

g. The sentence that begins "Ankle2 has previously..." in line 305 is required to cite a reference.

A reference has been added to this sentence.

Reviewer #2 (Remarks to the Author):

By using multiomics sequencing, the authors demonstrated that Zfp335 deficiency led to defective T cell development via dysregulation of the Zfp335/Ankle2/Baf axis, which ultimately drives cGAS/STING-mediated DN4 cell death. They have also suggested that DN4 cells are heterogeneous and defined a "true" DN4 cell population. However, while these findings are important and the conclusions are supported by solid data, a paper published very recently obtained similar conclusion that Zfp335 is an important regulator of early T cell development in the thymus via modulating apoptosis (Wang, Jiao, Sun, et al. eLife 2022;11:e75508). Thus, the novelty of the current study is a concern.

We thank the reviewer for their thoughtful comments and feedback. As for the novelty, we consider that our study presents different aspects and more details of the role for Zfp335 in T cell development, compared to the recent publication by Wang et al. For example, their study does not describe the mechanistic involvement of the cGAS/STING pathway during T cell development. Additionally, Wang et al., failed to provide strong evidence for the mechanism by which Zfp335 promotes T cell development. We show that Zfp335 functions to promote Ankle2 expression which in turn regulates Baf activity to ultimately prevent cGAS/STING-mediated cell death. Our study provides the first evidence for involvement of cGAS/STING in T cell development which we feel is the most novel aspect of this manuscript.

Questions:

1. What is the expression profile of Zfp335 and Ankle2 during different T cell development stages? whether Zfp335 deficiency also regulates apoptosis of DP thymocytes?

Related to Reviewer #1 comment 1, Zfp335 expression is ubiquitous throughout T cell development and peaks at the DP stage (Fig S1C). Similarly, Ankle2 is expressed throughout T cell development (Fig S8A). However, Ankle2 expression peaks at DN1 and DP stages.

Supplementary Figure 1C – Zfp335 expression across T cell development.

Supplementary Figure 8A – Ankle2 expression across T cell development

We also found that Zfp335 deficiency likely does not promote significant changes in DP apoptosis. Analysis of DP cells derived from 3 day OP9-DL1 culture revealed a slight but significant increase in frequency of Annexin V+ DP cell from Zfp335cKO mice. Since OP9-DL1 cells cannot support positive selection of DP cells this system is not ideal for assessing survival of DP thymocytes. Therefore, to determine if survival of DP cells was impaired *in vivo* we utilized TCR α repertoire analysis of total DP thymocytes. Since TCR α rearrangements progress from proximal V-proximal J to distal V-distal J in a time dependent manner assessment of gene usage in functional rearrangements could be used a surrogate for duration of DP survival [Carico et al., *Cell Reports*, (2017)]. As such, increased rates of Zfp335cKO DP apoptosis should yield a skewed TCR α repertoire towards proximal V-J pairings. No such skewing was observed. Indicating loss of Zfp335 does not lead to increased cell death among DP thymocytes *in vivo*. Additionally, assessment of positive selection showed no significant differences between WT and Zfp335cKO DP cells further supporting that loss of Zfp335 does not promote pre-mature apoptosis of DP thymocytes. These can be found in Supplementary Figure 4.

Supplementary Figure 4 – (A) % Annexin V+ among DP thymocytes derived from 3 day OP9-DL1 culture. Frequency of Trav (B) or Traj (C) gene segment usage in DP thymocyte TCRA rearrangements. Representative gating (D) and quantification (E) of positive selection among DP thymocytes.

2. Inactivation of pre-TCR signals dampens T-lymphocyte development by arresting thymocytes at the DN3 stage and inducing apoptosis (Fehling et al., 1995). Does Zfp335 deficiency regulate apoptosis via regulating pre-TCR activation?

Zfp335 deficiency does not appear to regulate apoptosis via regulating pre-TCR activation because we observe no deficiencies in either rearrangement or expression of pre-TCR or the ability of Zfp335cKO cells to transduce pre-TCR signals. These data can be found in Supplementary Figure 3.

- This was already shown in supplement 3

3. What determines the unique sensitivity of DN4 cells to STING-mediated apoptosis?

At this time the answer to this question is not entirely clear, however, we currently believe that the presence of TCR excision circles (TREC) may lead to increased basal cGAS/STING signaling in DN4 cells. We believe this basal cGAS/STING signaling reduces the signaling threshold for STING-mediated cell death in DN4 cells. We tested this hypothesis by generating Rag-deficient DN4 cells via *in vivo* αCD3 treatment and subjected them to STING agonist treatment *ex vivo*. These Rag-deficient DN4 cells did not exhibit increased sensitivity to STING agonist-induced apoptosis (Fig 7M). Additionally, we found that increasing the concentration of STING agonist 10-fold resulted in significantly increased cell death among most thymocyte subsets (Fig 7K) demonstrating an increased sensitivity to STING-mediated apoptosis in DN4 cells. The lack of sensitivity of DP cells to STING agonist-induced apoptosis is likely the result of strong down-regulation of cGAS and *STING* expression during the DN-DP transition and lack of cell division following Trac gene rearrangement thereby preventing the exposure of TRECs to cytosolic cGAS. We have added the following to the discussion on this topic:

"A surprising finding of our study was the unique sensitivity of DN4 thymocytes to cGAS/STING-mediated apoptosis. We did not observe significant increases in cell death among any other population of thymocytes following low-dose ex vivo STING activation. However, 10-fold higher STING agonist concentrations led to significant increases in cell death across all thymocyte subsets. This sensitivity was efficiently

abrogated upon *Bcl2* overexpression, suggesting a dominant role of pro-apoptotic members of this protein family in driving cell death. Interestingly, we found that preventing *V(D)J* recombination via deletion of *Rag2* was sufficient to protect DN4 cells from *STING*-mediated cell death suggesting that *V(D)J* recombination may play a role in this unique sensitivity to *cGAS/STING* signaling. A reasonable explanation for our finding is that *TRECs* generated by *V(D)J* recombination may become localized to the cytoplasm upon cell division at the DN4 stage, thereby providing activating signals for *cGAS*. This *cGAS* activity in turn may lower the signaling threshold for *STING*-mediated cell death. However, this does not explain the lack of DP cell sensitivity to *STING* agonist which undergo repeated rounds of *VJ* recombination. We propose that the apparent lack of sensitivity to *STING* signaling among DP cells is due to loss of *cGAS* and *STING* expression during the DN-DP transition (Immgen Data). Should this be true, it is reasonable to hypothesize that the down regulation of *cGAS* and *STING* expression is necessary to facilitate survival of DP thymocytes. Alternatively, the lack of cell division during T cell development beginning at *TCR α* recombination may prevent exposure of DP-derived *TRECs* to the cytosol. In either case, further study of this phenomena is warranted."

Figure 7 – (K) Frequency of apoptosis among WT thymocyte subsets stimulated with 250ug/mL CMA overnight. (M) Frequency of apoptosis among *Rag2*^{-/-} thymocyte subsets generated by *in vivo* administration of aCD3 antibody then stimulated with 25ug/mL CMA overnight.

4. There are 5 clusters of "True"DN4 cells. Whether Zfp335 plays different roles in apoptosis among these clusters?

At the gene signature level for apoptosis there are no differences between WT and Zfp335cKO DN4 cells (Fig S7J).

Supplementary Figure 7J – Violin plot of Reactome Apoptosis gene signature in WT (grey) and 'true' mutant Zfp335cKO (red) DN4 cells.

Reviewer #3 (Remarks to the Author):

The paper reports the role of transcription factor Zfp355 in the DN4 thymocyte development stage. I was tasked to review systems biology and computational biology aspects of the paper. While overall the computational analyses of the data look reasonable, I have major concerns that need to be resolved before publication (or in fact before I can reasonably assess the computational analyses).

Major comments:

- No data reported in the paper (bulk RNA-seq and scRNA-seq for WT and Zfp355-KO cells)

was provided. All the high-throughput data reported in the manuscript should be available for review in order to establish reproducibility and enable some validation. The paper states all data will be available at a certain GEO accession number, but currently is it not.

We thank Reviewer 3 for the comments and feedback. The RNA sequencing data presented in this manuscript has been submitted to GEO under accession GSE189244. The reviewer token for data access was previously provided in the reporting summary but for ease the token is: udsryeesvtejrml . New TCR repertoire sequencing data has been added to the manuscript and can be accessed through GSE205323 using the reviewer token: ufsfiwsmnlsjtab. To access the GEO profiles the accession number needs to be entered on the GEO website. It will not show up if you search for it using google or other search engines.

- Exposition of the data analysis in the main figures, figure captions, and in Methods are sometimes vague, and more detailed description of the methodology and content of the figures is warranted.

- I have also potential concerns / questions about some pieces of computational analysis where additional alternative analyses and/or parameters are needed to better establish robustness of observations.

Specifically:

- Fig 1J,K, Line 622: edgeR and DESeq2 were used for differential expression analysis. But how exactly? With which parameters? Typically the whole task of differential expression analysis is performed either entirely using edgeR, or entirely using DESeq2. It is not clear how the two methods were combined here.

Thank you for pointing this out. The computational pipeline we used for analysis implements edgeR for some pre-processing steps involving quality control, sample clustering and PCA. For differential expression analysis DESeq2 was exclusively used. The methods section has been updated to reflect this.

- Fig 1L: Same with GSEA analysis (Line 623). Not clear what parameters were used, and what exact lists of genes, coming from the previous step, were used. Not clear what exactly is shown in Fig 1L -- all enriched terms? Manually selected ones? Were all differentially expressed genes used for GSEA analysis or only the downregulated ones?

For GSEA default parameters were used with the MSigDb Hallmarks gene sets. Fig 1L shows the nominal enrichment scores for all pathways coming out of analysis with nominal p-values < 0.05. For the analysis all genes were included as a pre-ranked list generated using the following formula: $\text{rnk} = -\log_{10}(\text{p-val}) * (\text{fold-change} / |\text{fold-change}|)$.

- Fig 1J: caption says "differential expression of select Zfp355-target genes". Does this mean this differential expression analysis was restricted only to Zfp355 targets? If yes, then how were these targets defined? What is the motivation for this?

The differential expression analysis was not restricted to only Zfp335 target genes. The genes labelled were randomly selected from a list of genes identified by Zfp335 ChIP-seq from thymocytes. Not all targets are labelled as the plot would be unreadable. The figure legend has been updated to the following:

"(J) Volcano plot of differentially expressed genes between Zfp335cKO and WT by RNA-seq."

- Fig 1K: Some ChIP-seq data was used. But not clear how exactly it was analyzed. This is not explained in Methods.

Previously published ChIP-seq was used in this study. Details of exactly how the data were analyzed has been added to the methods section 'ChIP-seq Analysis'. Specifically, we added the following:

"Previously published E2A⁴⁰ (GSE89849) was generated by our lab or Zfp335⁴² (GSE58333) ChIP-seq datasets were accessed through the NCBI GEO database. Reads were aligned to the mm9 genome using Bowtie (version 1.1.2, parameters: -chunkmbs 128 -mm -m 1 -best -strata -p 4 -S -q). Peaks were called and Bed and wiggle files generated for visualization using the Integrative Genome Browser using MACS2 (version 1.4.2, default parameters). Peaks were annotated using the NGS: Peak Annotation tool on Nebula."

- The authors used cellranger, apparently with default parameters and standard mm10 mouse reference genome, to preprocess the data and obtain raw gene expression matrices. How accurate this is with respect to T cell receptor genes in their particular cells vs. these loci in the reference? How accurate are Trac, Trbc1, Tcfg-C1, etc. gene expression estimates obtained in this manner?

Recommended parameters for generation of reference genome and annotation files to be used in cellranger include annotation of all TCR gene segments. Based on the structure of TCR gene rearrangements and us having used 3' profiling assignment for V and J gene segments usage may not be entirely accurate since they are part of the 5' region of TCR transcripts. However, C regions of the TCR transcripts are 3' reducing the likelihood of failure to properly assign reads. Additionally, TCR genes are highly expressed making the likelihood of dropouts low. This is corroborated by the high rate of Trac and Trdc/Trgc detection in Mat and gd populations, respectively.

- Line 677: data was normalized, scaled, cell cycle scored -- how exactly and with which parameters?

We processed the scRNA-seq data according to standard processing pipelines implemented through Seurat using default parameters. Specifics of these default parameters have been added to the methods section. In short, count matrices were log normalized using a scale factor of 10,000 via the NormalizeData() function. Scaling was performed on all genes using a linear model via the ScaleData() function. Cell cycle scoring was performed using the S phase and G2M phase genes included in the Seurat software via the CellCycleScoring() function. The S and G2M phase genes can be accessed through Seurat via 'cc.genes'.

- Line 678: dropouts were imputed. Does this mean that the subsequent analysis uses imputed rather than raw expression values? It is controversial in the field if it is appropriate to rely on imputed scRNA-seq gene expression values for anything other than just visualizations of expression. Therefore I suggest that the authors repeat some of their major analytical steps without imputation, thus confirming their main results.

We are aware of the controversy surrounding the use of transformed data for scRNA-seq analysis. While we did perform dropout imputation it was only used for visualization and estimation of frequency of expression. All differential expression analyses were performed using the normalized count matrix and not dropout imputed data.

- Line 682: UMAP and clustering -- the text implies that clustering was performed after

UMAP, on UMAP-embedded (2-dimensional?) data. If this is the case, it is incorrect. Clustering should be performed on higher dimensional data, i.e. in this case, on 35-dimensional PCA. This should be clarified in the text of Methods. Also, line 683 refers to clustering with parameter 0.5, but what algorithm or Seurat function does this parameter refer to?

The authors apologize for any confusion. Clustering was performed following shared nearest neighbor graph generation using the first PCA dimensions but prior to UMAP embedding. Text in the methods has been updated to clarify this point. Line 683 states that "clustering was defined using a resolution of 0.5". The resolution parameter is used by the FindClusters() function of Seurat which utilizes Louvain algorithm to identify clusters. Adjustment of the resolution parameter controls the granularity of clustering with larger values returning a greater number of clusters.

- Fig 4C: How exactly were these frequencies calculated? i.e. frequencies of what in what are these values?

The frequencies in Fig 4C refer to the percentage of cell from WT or Zfp335cKO samples that fall into each cluster. Example calculation: Frequency of WT DN4_1 = (# WT DN_1 cells) / (total WT cells).

- Fig S5B,C: How exactly differential expression analysis was run in these plots?

For Fig S5B,C cells from the Zfp335cKO sample were defined as having high or low Zfp335 transcriptional activity based on the binning shown in Fig S5A. Cells falling in the targets high or targets low population were separately compared to all WT cells. Differential expression was performed using Wilcoxon tests on the normalized count matrix via the FindMarkers() function of Seurat. Genes expressed by less than 5% of cells or with log2 fold-change < 0.05 were excluded.

- Fig 5: caption doesn't correspond to figure. Probably plot correspond to caption (A) is not in the figure.

We thank the reviewer for pointing out this error. It has been resolved and the figure legend accurately updated.

REVIEWERS' COMMENTS

Reviewer #1 (Remarks to the Author):

All my concerns have been addressed properly.

Reviewer #2 (Remarks to the Author):

The authors have addressed most of my critiques and I have no further questions on this study.

Reviewer #3 (Remarks to the Author):

The authors adequately addressed my previous concerns, and I do not have any more major comments. To the best of my understanding, concerns of other reviewers were also properly addressed. I think the paper is now suitable for publication.

Minor comments:

- Fig 5D: please perform statistical analysis and report significance. For some plots, it's not even clear if the observation is that expression is about equal, or where it is higher.
- Fig 6B and 6C: numbers on top of bar plots are p-values? Is it coincidence or typo that they are identical?

RESPONSE TO REVIEWERS' COMMENTS

Reviewer #1 (Remarks to the Author):

All my concerns have been addressed properly.

The authors thank reviewer #1 for all their feedback during peer review of this manuscript.

Reviewer #2 (Remarks to the Author):

The authors have addressed most of my critiques and I have no further questions on this study.

The authors thank reviewer #2 for all their feedback during peer review of this manuscript.

Reviewer #3 (Remarks to the Author):

The authors adequately addressed my previous concerns, and I do not have any more major comments. To the best of my understanding, concerns of other reviewers were also properly addressed. I think the paper is now suitable for publication.

The authors thank reviewer #3 for all their feedback during peer review of this manuscript.

Minor comments:

- Fig 5D: please perform statistical analysis and report significance. For some plots, it's not even clear if the observation is that expression is about equal, or where it is higher.

We have added log₂FC and p-values for the comparison of expression between WT and Zfp335cKO cells in Fig 5d to each violin plot.

- Fig 6B and 6C: numbers on top of bar plots are p-values? Is it coincidence or typo that they are identical?

The numbers on top of the bar plots are p-values. It is not a coincidence or typo that the values are identical. For these experiments we could not satisfy the assumptions necessary to perform parametric statistical tests and therefore utilized non-parametric Mann-Whitney U tests. P-values calculated by Mann-Whitney U tests are only dependent upon the total sample size between both groups and the rank of datapoints between each group. Since there are no overlapping values between the two groups for either graph and both graphs have the same sample size the p-values for both comparisons will always be identical.